# SGD-SM 2.0: An Improved Seamless Global Daily Soil Moisture Long-term Dataset From 2002 to 2022

Qiang Zhang[1], Qiangqiang Yuan[2][*], Taoyong Jin[2][*], Meiping Song[1], and Fujun Sun[3]

[1]Center for Hyperspectral Imaging in Remote Sensing (CHIRS), Information Science and Technology College, Dalian Maritime University, China

[2]School of Geodesy and Geomatics, Wuhan University, China

[3]CASIC Research Institute of Intelligent Decision Engineering, Beijing, China

**Correspondence:** Qiangqiang Yuan (yqiang86@gmail.com) and Taoyong Jin (tyjin@sgg.whu.edu.cn)

**Abstract.** Satellite-based daily soil moisture products inevitably exist the drawbacks of low-coverage rate in global land, because of the satellite orbit covering scopes and the limitations of soil moisture retrieving models. To solve this issue, Zhang et al. (2021a) generated seamless global daily soil moisture (SGD-SM 1.0) products for the years 2013∼2019. Nevertheless, there are still several shortages in SGD-SM 1.0 products, especially on temporal range, sudden extreme weather condition, and sequen-

tial time-series information. In this work, we develop an improved seamless global daily soil moisture (SGD-SM 2.0) dataset from 2002 to 2022, to overcome above shortages. SGD-SM 2.0 uses three sensors AMSR-E, AMSR2 and WindSat. Global daily precipitation products are fused into the proposed reconstructing model. We propose an integrated long and short-term memory convolutional neural network (LSTM-CNN) to fill the gaps and missing regions in daily soil moisture products. In-situ validation and time-series validation testify the reconstructing accuracy and availability of SGD-SM 2.0 (R: 0.672, RMSE:

0.096, MAE: 0.078). The time-series curves of the improved SGD-SM 2.0 are consistency with the original daily time-series soil moisture and precipitation distribution. Compared with SGD-SM 1.0, the improved SGD-SM 2.0 outperforms on reconstructing accuracy and time-series consistency. SGD-SM 2.0 products are recorded at **https://doi.org/10.5281/zenodo.6041561** (Zhang et al., 2022).

## 1   Introduction

Surface soil moisture acts as a significant part on global hydrology and meteorology, especially for forecasting drought and flood disasters (Wigneron et al., 1999; Long et al., 2014; Brocca et al., 2018). In recent years, satellite-based soil moisture retrieving data has been rapidly progressed on both global and daily monitoring (Shi et al., 2006; Dorigo et al., 2012; Al Bitar et al., 2017; Dorigo et al., 2021). For example, AMSR-E, AMSR2, WindSat global daily soil moisture products and so on (Fan et al., 2004). These quantitative products have been widely utilized for global and long-term hydrological analysis and forecast

(Chen et al., 2021; Todd-Brown et al., 2021).

    However, because of the limitations of soil moisture retrieving models and satellite orbital covering scopes, the obtained daily soil moisture products are fragmentary and incomplete (Shi et al., 2002; Enenkel et al, 2016; Meng et al., 2021). As

shown in Fig. 1(a) and (b), these soil moisture products exist plenty of gap regions. Actually, the land coverage rate is only approximately 20% to 80% in daily AMSR-E/2 and WindSat quantitative products (Long et al., 2019).

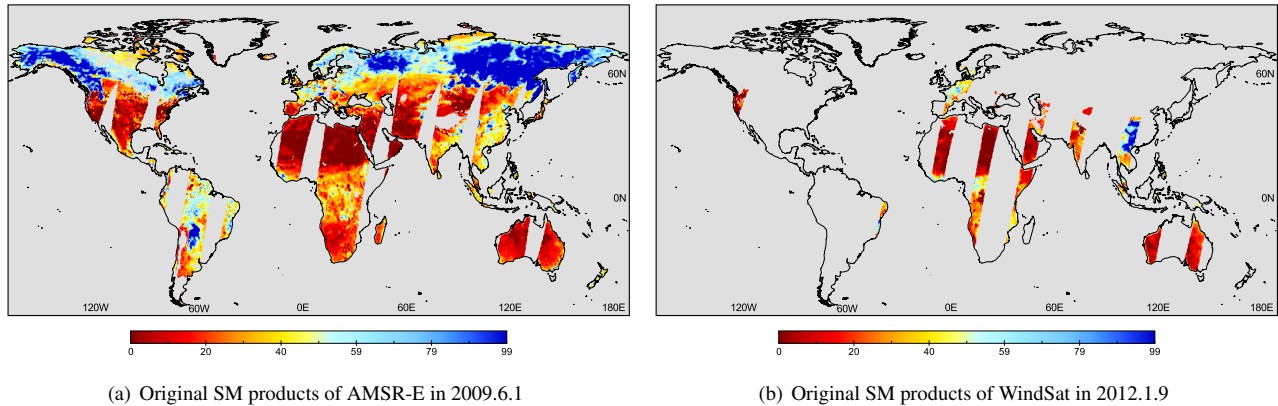

(a) Original SM products of AMSR-E in 2009.6.1          (b) Original SM products of WindSat in 2012.1.9

**Figure 1.** Daily soil moisture products of AMSR-E and WindSat.

To settle this adverse effect for global soil moisture applications, most of works adopted the temporal averaging operation such as monthly, quarterly, or yearly averaging (Schaffitel et al., 2020; Guevara et al., 2021; Wang et al., 2021). This strategy could usually acquire full-coverage soil moisture products via averaging abundant daily products. Nevertheless, temporal averaging operation is also a two-edged sword. Firstly, it directly replaces daily temporal resolution with low-frequency temporal resolution (Rebel et al., 2012; Long et al., 2020), which greatly lowers the utilization of daily soil moisture products. Secondly,

temporal averaging operation disregards the specific spatial distribution of daily products, and neglects the sequential time-series changing characteristic (Zeng et al., 2015a; Wang et al., 2021). In other words, monthly, quarterly, or yearly averaging strategy degrades the original characteristics for daily soil moisture products.

     To address this issue, Zhang et al. (2021a) generated a seamless, global, daily soil moisture (named SGD-SM 1.0) dataset from 2013 to 2019. The spatial resolution is denoted as 0.25° (about 25km). SGD-SM 1.0 relies on the deep spatio-temporal

partial convolutional model to fill the gaps or missing regions in daily soil moisture products. Then three validations are performed to verify the reliability of SGD-SM 1.0 products. Relevant quantitative indexes (R, RMSE and MAE) and results demonstrate that SGD-SM 1.0 products can be extended for global, daily and full-coverage soil moisture measurements (Zhang et al., 2021a).

     SGD-SM 1.0 maintains the original high-frequency daily temporal-resolution, and effectively enhances the utilization of

global daily soil moisture products. However, SGD-SM 1.0 also exists several weaknesses and limitations. Based on SGD-SM 1.0 and above considerations, we develop an improved seamless global daily soil moisture (SGD-SM 2.0) dataset for the years 2002-2022 in this work. Compared with SGD-SM 1.0, the main improvements and contributions of SGD-SM 2.0 are listed as follows:

★ SGD-SM 1.0 only uses single sensor (AMSR2), and the temporal range is insufficient with just seven years. While global soil moisture analysis and applications generally need longer-term and more multi-sensors products. The application range of SGD-SM 1.0 is still limited. Compared with SGD-SM 1.0, SGD-SM 2.0 uses three passive microwave sensors (AMSR-E, WindSat, and AMSR2). Temporal range of SGD-SM 2.0 is extended to twenty years from 2002 to 2022. The application scope of SGD-SM 2.0 could be enlarged through these long-term soil moisture products.

★ SGD-SM 1.0 ignores the daily extreme weather condition. If one day occurs a sudden precipitation, SGD-SM 1.0 usually performs poor under this scenario. The main reason is that SGD-SM 1.0 relies on the internal spatio-temporal correlation, which not considers the external environmental factors. Compared with SGD-SM 1.0, SGD-SM 2.0 introduces the global daily precipitation products into the reconstructing framework. Through fusing auxiliary precipitation data, SGD-SM 2.0 could lead in the daily extreme weather information for gap-filling.

★ Although SGD-SM 1.0 employs 3-D partial convolutional neural network to exploit both spatial and temporal feature, it is still insufficient for utilizing sequential time-series information. For daily soil moisture products, how to effectively reconstruct gaps missing regions through interrelated temporal information is significant. Compared with SGD-SM 1.0, SGD-SM 2.0 develops an integrated long and short-term memory convolutional neural network (LSTM-CNN) to fill the gaps and missing regions in these daily products. The proposed LSTM-CNN model could simultaneously utilize recurrent time-series information and spatial information.

★ Compared with SGD-SM 1.0 products, SGD-SM 2.0 products outperform on R (0.688), RMSE (0.094), and MAE (0.077). In addition, the time-series curves of the improved SGD-SM 2.0 products are more consistency with the original daily time-series soil moisture values. Benefiting from the data fusion of daily precipitation information, the proposed LSTM module can extract time-series features for filling the gaps and missing regions in daily soil moisture products. Therefore, SGD-SM 2.0 can be effectively utilized for global hydrology monitoring analyzing at fine (daily) temporal resolution.

The outline of this paper is arranged below. Sect. 2 provides a description of products and data used in this work. Sect. 3 gives the methodology of the proposed reconstructing framework for SGD-SM 2.0. Sect. 4 lists the experimental results of SGD-SM 2.0 products. Sect. 5 discusses the comparisons between SGD-SM 1.0 and SGD-SM 2.0, especially on reconstructing accuracy and time-series consistency. Finally, the conclusion and outlook are summarized in Sect. 6.

## 2 Products and data description

In this work, we simultaneously fuse satellite-based soil moisture products and precipitation products to generate SGD-SM 2.0 dataset. The in-situ soil moisture sites are employed to validate the reconstructing precision of SGD-SM 2.0. These in-situ data are downloaded from International Soil Moisture Network (ISMN). Detailed descriptions are listed as follows.

## 2.1 Satellite-based soil moisture products

AMSR-E/2 and WindSat global daily soil moisture products are utilized from 2002 to 2022. These three sensors are onboarded at Aqua satellite, GCOM-W1 and Coriolis satellite, respectively (Nepal et al., 2021). AMSR-E, AMSR2 and WindSat are all passive sensors for soil moisture retrieving. The spatial resolution is all 0.25° grid (about 25km) in these products, as depicted in Fig. 1(a)-(c). The retrieving model adopts the land parameter retrieval model (LPRM) for AMSR-E, WindSat, and AMSR2 products (McColl et al., 2017). We select the descending orbit (night-time), and 6.9 GHz band for all these soil moisture products. These datasets are all recorded at GES DISC website (NASA GES DISC, 2022). These three products provide the original information for the using of SGD-SM 2.0. The proposed reconstructing model acquires the gap masks and relies on the valid spatio-temporal soil moisture information from these three products, to fill the missing and gap regions.

The time-series range of AMSR-E sensor starts from 2002.06.19 and ends to 2011.10.04 (Njoku et al., 2003; Shi et al., 2008). The time-series range of WindSat sensor starts from 2003.02.01 and ends to 2012.08.02. The time-series range of AMSR2 sensor starts from 2012.07.03 and continues to current date (Zeng et al., 2020). In consideration of the low-coverage rate in WindSat dataset, we just use WindSat global daily products from 2011.10.5 to 2012.07.02, for acquiring sequential daily products. These recorded AMSR-E, WindSat and AMSR2 global daily products are all employed as the initial input of the proposed LSTM-CNN model for generating SGD-SM 2.0 products. The daily coverage rate curves of these three global quantitative products are depicted in Fig. 2(a)-(c), respectively.

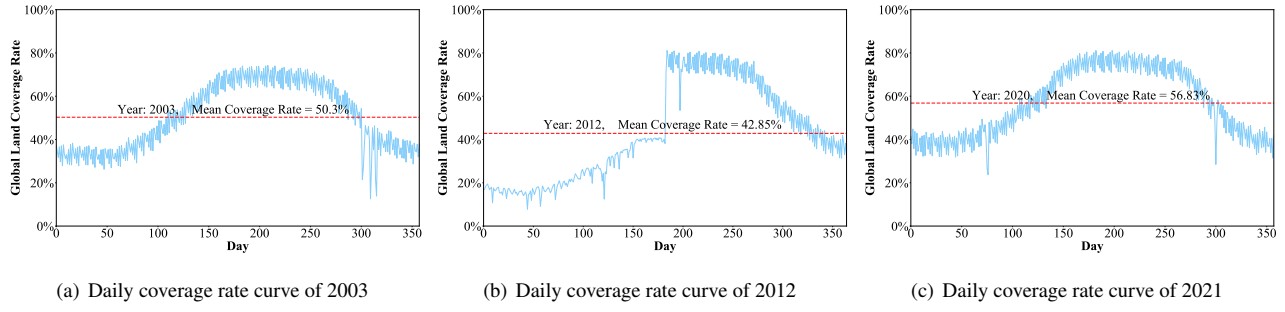

(a) Daily coverage rate curve of 2003  (b) Daily coverage rate curve of 2012  (c) Daily coverage rate curve of 2021

**Figure 2.** Daily coverage rate curves of AMSR-E, WindSat, and AMSR2 soil moisture products in 2002, 2012, and 2021.

## 2.2 Precipitation products

Precipitation usually has a high correlation with soil moisture in the corresponding regions (Pellarin et al., 2009; Brocca et al., 2014; Sun and Fu, 2021). Therefore, we fuse the precipitation products into the proposed SGD-SM 2.0 dataset to improve the reconstructing accuracy. The Integrated Multi-satellitE Retrievals for GPM (IMERG) global daily precipitation V6 products are employed for the years 2002∼2022 (Massari et al., 2020). These precipitation products are derived from multiple precipitation-relevant satellite passive microwave sensors, as portrayed in Fig. 3(a). The spatial resolution denotes as 0.1° grid (about 10km) in IMERG level 3 global daily final precipitation products. To keep the uniformity with soil moisture

products, the spatial downsampling operation is carried out for the original IMERG precipitation products from 0.1° to 0.25°. Then we normalize these precipitation values via linear transformation for the use of reconstructing model. These precipitation products were all downloaded from GES DISC (Brocca et al., 2019; Berg et al., 2021; Škrk et al., 2021).

### 2.3 In-situ soil moisture data

In-situ soil moisture sites are significant for testifying the satellite-based products (Brocca et al., 2014; Gruber et al., 2020). These sites provide high-precision surface soil moisture values. Relied on in-situ data, the quantitative indexes could be derived for the proposed SGD-SM 2.0 dataset. ISMN unites global in-situ surface data, which has been widely applied for hydrology and soil moisture validation (Dorigo et al., 2011; Wigneron et al., 2013; Dorigo et al., 2013; Dorigo et al., 2021). We select 124 stations from ISMN from 2002 to 2022 and match them with corresponding soil moisture product in SGD-SM 2.0 (Zhang et al., 2020). The selected criteria include three points: 1) The in-situ soil moisture sites are downloadable through the given website. 2) The in-situ soil moisture sites are continuous for the long-term observation, at least one year. 3) The spatial distribution of these in-situ sites covers various continents, land use and soil types. The spatial distribution of these selected in-situ data is displayed in Fig. 3(b). These in-situ soil moisture data are public and could be downloaded at https://ismn.geo.tuwien.ac.at/en/. The in-situ validation results of SGD-SM 2.0 and the reconstructing accuracy comparisons with SGD-SM 1.0 are given in section 4.2 and section 5.1, respectively.

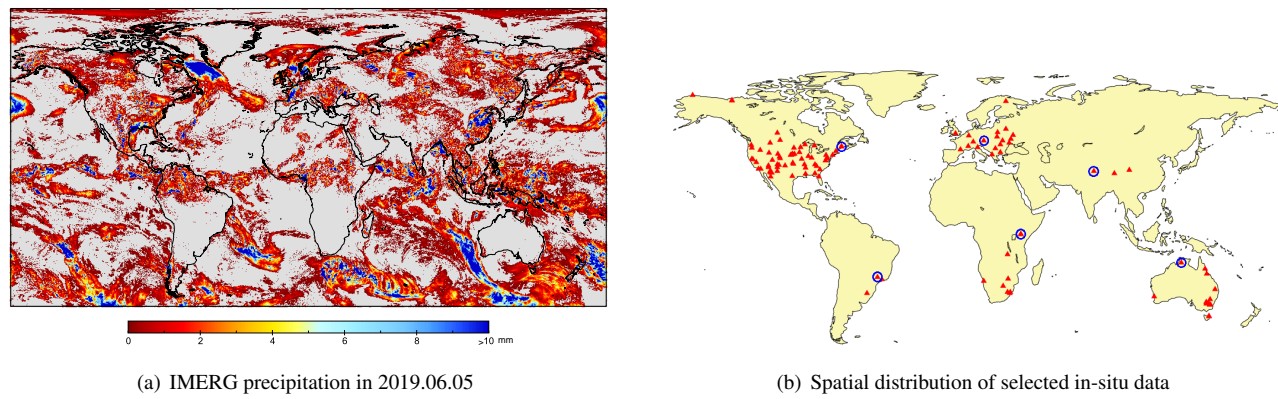

(a)  IMERG precipitation in 2019.06.05                     (b)  Spatial distribution of selected in-situ data

**Figure 3.** IMERG global daily precipitation and selected in-situ data.

### 3  Methodology

The schematic of the proposed work is depicted in Fig. 4. Different from SGD-SM 1.0, we simultaneously fuse global daily precipitation products with global daily soil moisture products into SGD-SM 2.0. An integrated long and short-term memory convolutional neural network (LSTM-CNN) reconstructing model is developed to fill the gap and missing regions in global

daily soil moisture products. Finally, we recursively generate the seamless daily soil moisture products in SGD-SM 2.0 dataset. Detailed descriptions of the proposed LSTM-CNN reconstructing model, training and optimization are stated below.

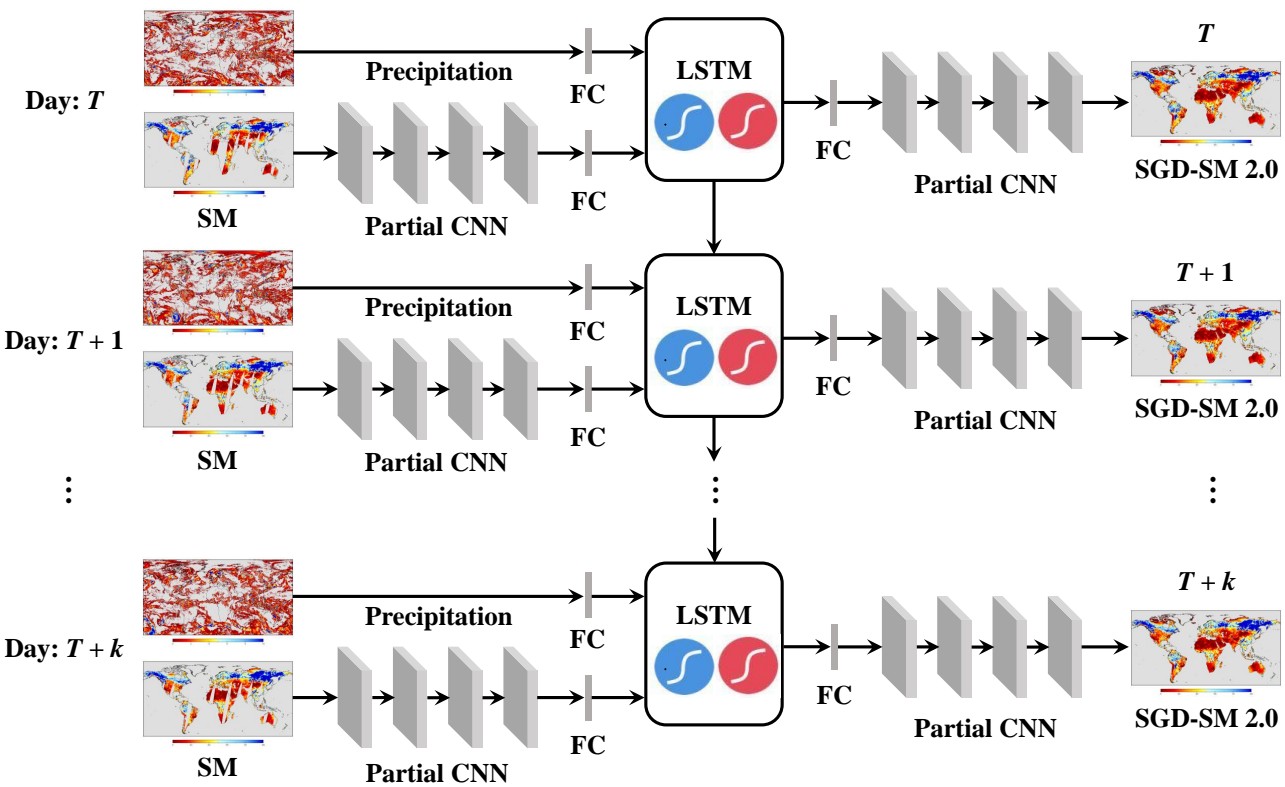

**Figure 4.** Schematic of the proposed framework to generate SGD-SM 2.0 products.

## 3.1 LSTM-CNN reconstructing model

As shown in Fig. 4, original global daily soil moisture product in date $T$ (AMSR-E, WindSat or AMSR2) and its corre-
120 sponding global daily precipitation product are utilized as the input data of the proposed framework. Firstly, the precipitation data in date $T$ is transformed as the vector value $P_T$ through a full-connected (FC in Fig. 4) layer. We employ the partial convolutional neural network (Partial CNN in Fig. 4) to extract the spatial feature of soil moisture product in date $T$. Different from the common CNN (Yuan et al., 2019), partial CNN can effectively acquire the spatial information within valid regions, and eliminate the invalid information within gap or soil moisture missing regions (Zhang et al., 2018a). We applied partial
CNN for generating SGD-SM 1.0 dataset. Due to its effectiveness on incomplete soil moisture products, the partial CNN is also used in this work for generating SGD-SM 2.0 dataset. The formula of partial CNN in this work is determined as follow:

$$\mathbf{S}'_{(m,n)} = \begin{cases} \mathbf{W}^\top (\mathbf{S}_{(m,n)} \otimes \mathbf{M}_{(m,n)}) \frac{\left\| \mathbf{1}_{(m,n)} \right\|_1}{\left\| \mathbf{M}_{(m,n)} \right\|_1} + b, & \left\| \mathbf{M}_{(m,n)} \right\|_1 \neq 0 \\ 0, & otherwise \end{cases} \tag{1}$$

where $\mathbf{M}$ denotes the mask of its corresponding soil moisture product $\mathbf{S}$. 0 and 1 refer to the invalid and valid point in mask $\mathbf{M}$. $\mathbf{W}$ and $b$ stand for trainable weighted and offset arguments in partial CNN, respectively. $\otimes$ represents the dot product operation, to exclude the invalid information in gap or missing regions through mask data. Subsequently, current mask $\mathbf{M}$ needs to regenerated under the below paradigm: If the partial convolution can generate at least one valid value of the output result, we need mark this location as valid value in the new masks. The mask regenerated formula is defined as follow (Zhang et al., 2020):

$$\mathbf{M}'_{(m,n)} = \begin{cases} \mathbf{L}_{(m,n)}, & \left\| \mathbf{M}_{(m,n)} \right\|_1 \neq 0 \\ 0, & other \end{cases} \tag{2}$$

where $\mathbf{L}_{(m,n)}$ stand for Earth land in position $(m, n)$. It should be noted that the Earth land mask includes 6 continents and neglects all regions of Antarctica and most regions of Greenland. The main reason is that these omitted regions are perennially covered with snowy or frozen land (Zhao et al., 2021).

After four partial CNN layers in Fig. 4, an FC layer is also acted on the feature maps of soil moisture product, with the result of vector value $S_T$:

$$S_T = \mathrm{FC}(\mathbf{S}'_4) \ s.t. \ P_T = \mathrm{FC}(\mathbf{P}_T) \tag{3}$$

Then the two vectors $S_T$ and $P_T$ of soil moisture and precipitation products are simultaneously imported into the LSTM module in Fig. 4. The architecture of the LSTM module within the proposed framework is displayed in Fig. 5.

As depicted in Fig. 5, soil moisture information $S_T$, corresponding precipitation information $P_T$, previous long-term memory information $C_{T-1}$, and previous short-term memory information $h_{T-1}$ are simultaneously imported into the LSTM module. The output values in LSTM are the regenerated soil moisture information $S'_T$, current long-term memory information $C_T$, and current short-term memory information $h_T$. It should be noted that, current long and short-term memory information in date $T$ is the previous long and short-term memory information in next date $T+1$, respectively. For memory information $h_0$ and $C_0$, these vectors are initialized with zero elements. LSTM is composed of three gates: Oblivious gate, input gate and output gate to control memory information state (Zhang et al., 2021b).

*1) Oblivious gate:* This gate determines which information needs to be discarded in the short-term memory state. It is carried out by the sigmoid unit $\sigma$ between the soil moisture information $S_T$ and previous long-term memory information $C_{T-1}$.

$$f_T = \sigma(W_f \cdot [h_{T-1}, S_T] + b_f) \tag{4}$$

where the sigmoid unit $\sigma$ is defined below:

$$\sigma(a) = \frac{1}{1 + e^{-a}} \tag{5}$$

In the sigmoid unit $\sigma$, zero means fail and one means pass for current information. Through checking the soil moisture information $S_T$ and previous long-term memory information $C_{T-1}$, it generates a vector between 0 and 1. This variable determines which information is retained or discarded in the short-term memory state.

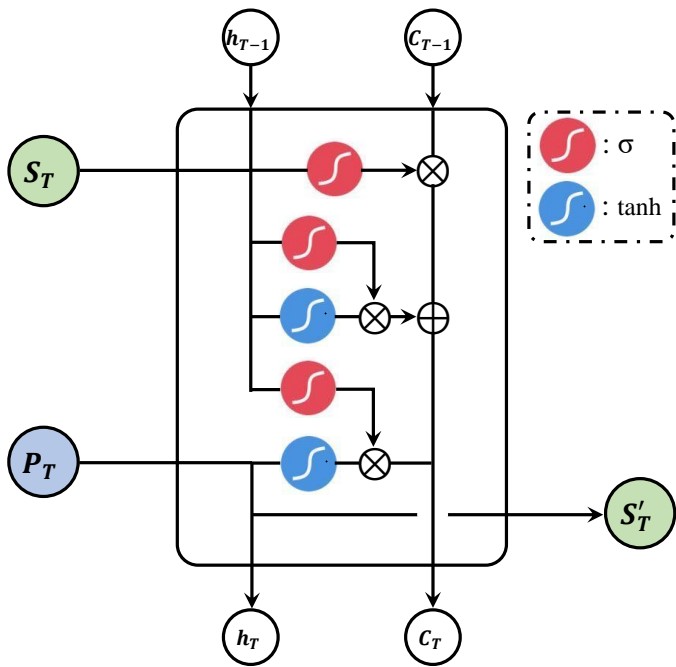

**Figure 5.** Structure of the LSTM module in the proposed framework.

*2) Input gate:* This gate determines what new information is added to the long-term memory state. Firstly, we use $h_{T-1}$ and $S_T$ to determine which information needs to be updated through the sigmoid operation in Eq. (6):

$$i_T = \sigma(W_i \cdot [h_{T-1}, S_T] + b_i) \tag{6}$$

Then the new candidate long-term memory information $\tilde{C}_T$ is generated through the tanh unit in Eq. (7):

$$\tilde{C}_T = \tanh(W_C \cdot [h_{T-1}, S_T] + b_C) \tag{7}$$

where the tanh unit is defined as:

$$\tanh(a) = \frac{e^a - e^{-a}}{e^a + e^{-a}} \tag{8}$$

*3) Output gate:* In this gate, we need to output current long-term memory information $C_T$ from previous $C_{T-1}$ and candidate long-term memory information $\tilde{C}_T$:

$$C_T = f_T \otimes C_{T-1} + i_T \otimes \tilde{C}_T \tag{9}$$

After updating current long-term memory information $C_T$, the regenerated soil moisture information $S'_T$ is output through previous short-term memory $h_{T-1}$, soil moisture information $S_T$, and corresponding precipitation information $P_T$:

$$S'_T = \sigma(W_{S'}[h_{T-1}, S_T] + b_{S'}) \otimes \tanh(P_T) \tag{10}$$

Besides, we need to output current short-term memory information $h_T$ for the next date $T+1$ as follow:

$$h_T = \tanh(o_T \otimes C_T) \tag{11}$$

Later, the regenerated soil moisture information $S'_T$ is transformed by a FC layer in the right of Fig. 4. Then four partial CNN layers are performed, to generate the final SGD-SM 2.0 product in date $T$. Through the consecutive time-series strategy, we recursively reconstruct the daily soil moisture products in SGD-SM 2.0.

## 3.2 Training and optimization

To generate reliable and high-precision SGD-SM 2.0 dataset, how to train and optimize the proposed LSTM-CNN model is extremely crucial in this work. The training stage needs huge numbers of sample labels, to optimize the trainable parameters in the proposed partial CNN and LSTM in Fig. 4 and Fig. 5, respectively. The sample labels adopt patch selecting strategy. We select sequential time-series daily soil moisture patches with $k = 7$ in the reconstructing framework. The spatial size of these seven-day soil moisture patches is all set as 40×40. These time-series seven-day soil moisture patches are all complete, without gap or data missing regions from the original soil moisture products. Then, we randomly select 30000 mask patches with the spatial size of 40×40. Each soil moisture patch is simulated with missing regions via these mask patches. Through this way, we acquire 30000 training samples from the original 2002-2022 soil moisture products. Each training sample includes four variables: the simulated seven-day soil moisture patches, the complete seven-day soil moisture patches, the corresponding mask patches, and the corresponding precipitation patches. These variables are simultaneously imported into the LSTM-CNN reconstructing model, as shown in Fig. 4.

For the partial CNN in the proposed framework, we set the convolutional filter size as 3×3 in all the partial CNN layers (Xiao et al., 2022a). The last partial CNN layer outputs just one feature map and the other partial CNN layers output 64 feature maps (Xiao et al., 2022b). ReLU is utilized after each partial CNN layer. For the LSTM module in the proposed framework, we set the dimension of long and short-term memory vectors $C_T$ and $h_T$ as 2048.

For the network optimization, we adopt the same strategy with the global-local function (Zhang et al., 2021a) in SGD-SM 1.0. The global soil moisture uniformity and local soil moisture heterogeneity are both taken into consideration in the proposed LSTM-CNN reconstructing model. Different from SGD-SM 1.0, we simultaneously fill the gap and missing regions in time-series seven-day soil moisture patches. Detailed definitions of the global-local function are determined as follows:

$$\xi(\mathbf{W}, b, W_{f,i,C,S'}, b_{f,i,C,S'}) = \sum_{T=1}^{k} \left( \left\| (1 - \mathbf{M}_T) \otimes (\mathbf{S}_T^{rec} - \mathbf{S}_T^{ori}) \right\|_2^2 + \left\| (\mathbf{M}_L \otimes (\mathbf{S}_T^{rec} - \mathbf{S}_T^{ori}) \right\|_2^2 \right) \tag{12}$$

where $\mathbf{M}_L$ represents the global land mask (including 6 continents and neglecting all regions of Antarctica and most regions of Greenland). $\alpha$ stands for the balancing parameter to equilibrate the local loss and global loss (Zhang et al., 2020). Empirically, this ratio is fixed as 0.1 in the training and optimization stage.

In terms of the hyper-parameters and operations of the proposed framework, related explanations are listed below. The batch size of the LSTM-CNN reconstructing model is set as 128 (Zhang et al., 2018a). The whole epoch number is confirmed

determined as 500 (One epoch represents that all the samples in the training set have been utilized for the neural network optimization at one time). The inceptive learning rate is started as 0.005 (Zhang et al., 2018b). It gradually decreases through multiplying a damping factor (equal to 0.5) every 100 epochs (Zhang et al., 2019). On software configuration, LSTM-CNN

model is carried out on PyTorch 1.8.1 framework. We use Python 3.7 language, PyCharm platform, and Windows 10 environment to generate seamless global daily soil moisture products. On hardware configuration, we employ a NVIDIA Titan X (Pascal) GPU, Inter E5-2609v3 CPU, and 16 GB DDR4 RAM to execute the proposed LSTM-CNN model.

## 4 Experiments and validations

The released SGD-SM 2.0 products are recorded at **https://doi.org/10.5281/zenodo.6041561** (Zhang et al., 2022). SGD-

210 SM 2.0 starts from 2002.06.23, and ends at 2022.02.05. The initial and reconstructing global daily soil moisture products have been stored with individual NetCDF4 (*.nc) document. Because part of daily soil moisture products is missing at GES DISC, these products are also neglected in the proposed SGD-SM 2.0 dataset (7115 files). In this section, the experimental results of SGD-SM 2.0 dataset are given in section 4.1. Later, we carry out the in-situ validation and time-series validation of SGD-SM 2.0 in section 4.2 and section 4.3, respectively.

### 4.1 Experimental results

As shown in Fig. 6 and Fig. 7, the SM and SGD-SM 2.0 results are given in 10, 20, and 30 September 2002 and in 10, 20, and 30 June 2020, respectively.

For comparison purpose, the left lines are the original global daily products and the right lines are the reconstructed SGD-SM 2.0 products in Fig. 6 and Fig. 7. It should be noted that we neglect all regions in Antarctica and most regions in Greenland,

because of the perpetual frozen soil. Clearly, gaps and missing regions are filled through the proposed framework in Sect. 3.

From the spatial perspective, the proposed SGD-SM 2.0 dataset performs both global soil moisture uniformity and local soil moisture heterogeneity in Fig. 6 and Fig. 7. It ensures the spatial consistency especially for the gap regions with the adjoin soil moisture regions. Beyond that, the reconstructed regions in SGD-SM 2.0 don't reflect distinct patch or border effect. This also testifies the powerful ability of partial CNN in the proposed framework, which can effectively exclude the invalid information

in gap or missing soil moisture regions.

From the temporal perspective, the proposed SGD-SM 2.0 dataset utilizes the complementary and sequential time-series soil moisture information. Through fusing global daily precipitation products, SGD-SM 2.0 can consider the sporadic extreme weather condition for single day. In addition, by means of LSTM module, the consistent temporal information can be recovered and preserved in Fig. 6 and Fig. 7.

### 4.2 In-situ validation

In-situ validation is the most reliable method to measure the accuracy and availability of the proposed SGD-SM 2.0 dataset (Walker et al., 2004; Draper et al., 2009; Zeng et al., 2015b). In this work, we choose 124 in-situ surface (0∼5cm depth) soil

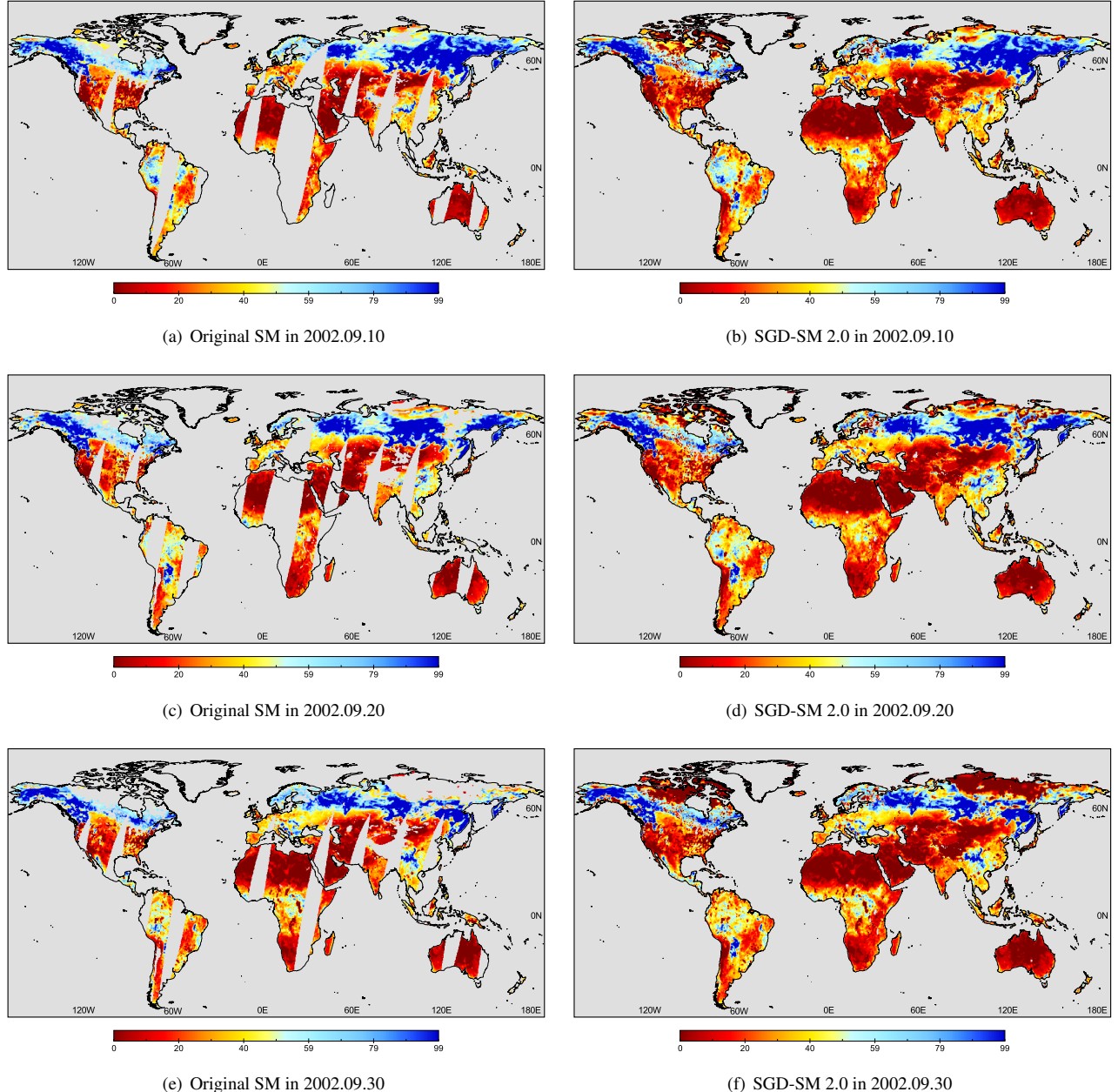

(a) Original SM in 2002.09.10

(b) SGD-SM 2.0 in 2002.09.10

(c) Original SM in 2002.09.20

(d) SGD-SM 2.0 in 2002.09.20

(e) Original SM in 2002.09.30

(f) SGD-SM 2.0 in 2002.09.30

**Figure 6.** Original SM and proposed SGD-SM 2.0 results in 10, 20, and 30 September 2002.

moisture sites from ISMN, as shown in Fig. 3(b). The selected in-site values are limited from 2002 to 2022. We match the hourly in-site values with the descending products. In consideration of validation reliability, we choose the two neighboring
in-site values correspond with the observation time of soil moisture products. Then we average them as the ground-truth data.

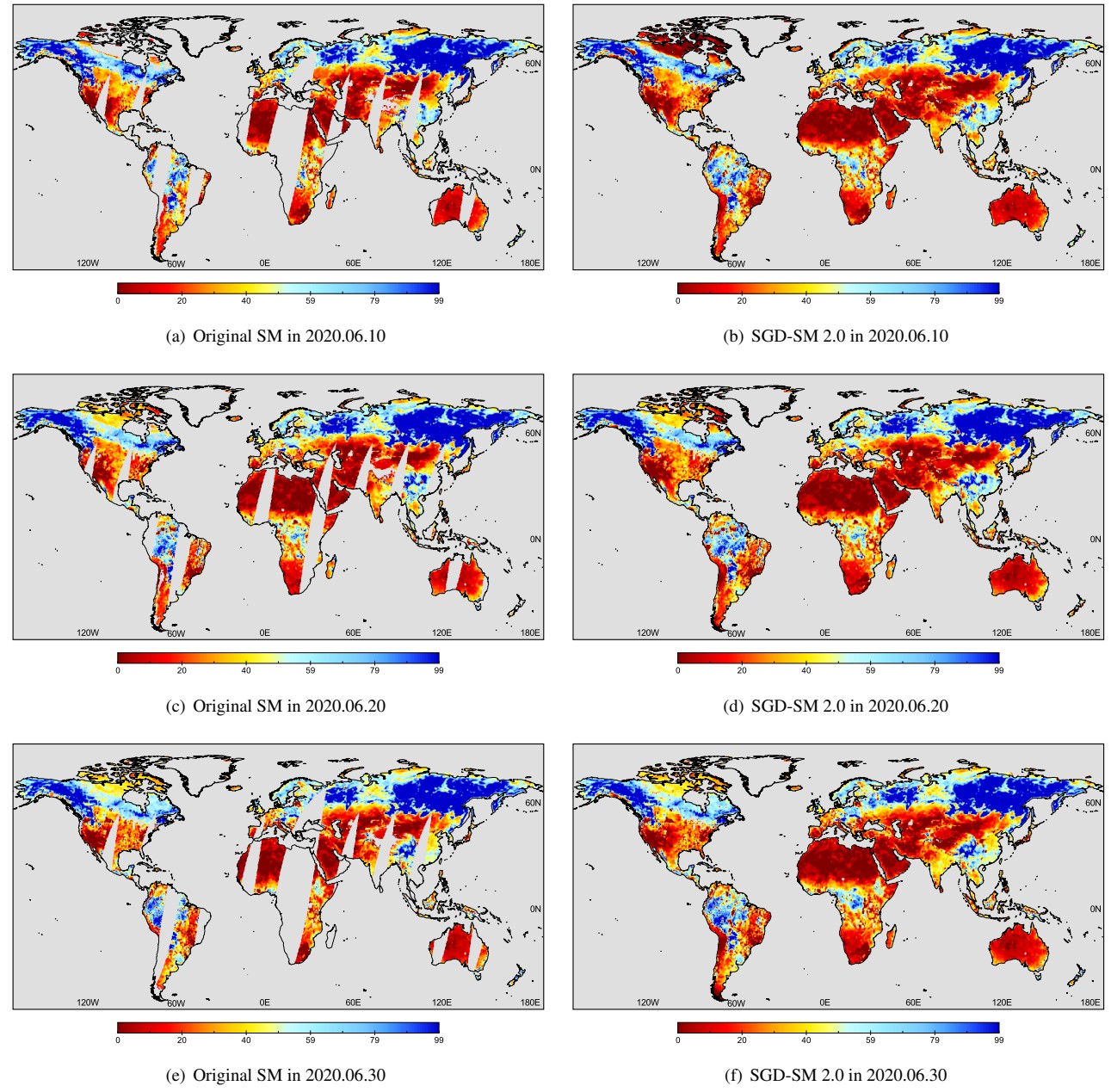

(a) Original SM in 2020.06.10

(b) SGD-SM 2.0 in 2020.06.10

(c) Original SM in 2020.06.20

(d) SGD-SM 2.0 in 2020.06.20

(e) Original SM in 2020.06.30

(f) SGD-SM 2.0 in 2020.06.30

**Figure 7.** Original SM and proposed SGD-SM 2.0 results in 10, 20, and 30 June 2020.

As portrayed in Fig. 8, the scatters of six in-situ soil moisture sites (marked as blue circles in Fig. 3b: 42.537°N, 72.171°W; 0.282°N, 36.866°E; 48.141°N, 15.171°E; 14.159°S, 131.388°E; 21.617°S, 47.632°W; 31.369°N, 91.899°E) are displayed to demonstrate the reconstructing accuracy of SGD-SM 2.0. The horizontal coordinate refers to in-situ data. Accordingly, the

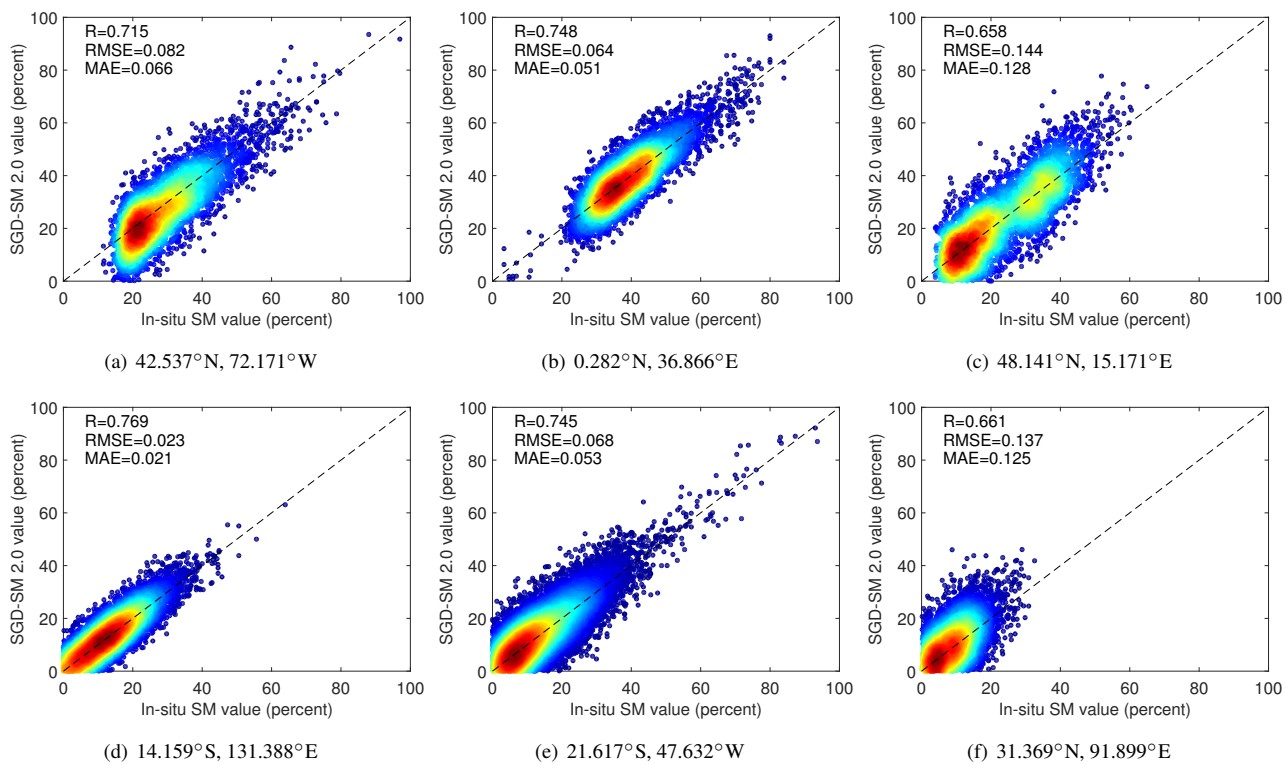

**Figure 8.** Scatters of six in-situ sites (Horizontal coordinate refers to the in-situ data; Vertical coordinate denotes the reconstructing data).

vertical coordinate denotes reconstructing data in gaps or missing soil moisture regions. The time range is limited from 2002 to 240 2022. The R indicators of these sites are varied from 0.658 to 0.769 in Fig. 8(a)-(f). The RMSE indicators and MAE indicators of these sites are varied from 0.023 to 0.144 and from 0.021 to 0.128 in Fig. 8(a)-(f), respectively.

**Table 1.** Comparisons between the original and SGD-SM 2.0 products (from 2002 to 2022) through 124 selected in-situ sites.

| Soil moisture products (2002∼2022) | Average evaluation indicators | | | |
|:---:|:---:|:---:|:---:|:---:|
| | R | RMSE | ubRMSE | MAE |
| Original | 0.679 | 0.094 | 0.058 | 0.075 |
| SGD-SM 2.0 | 0.672 | 0.096 | 0.061 | 0.078 |

Through all the 124 selected in-situ sites, Table 1 compares the original products with SGD-SM 2.0. The average evaluation indicators (R, RMSE, and MAE) of original soil moisture and SGD-SM 2.0 products are 0.679 (0.672), 0.094 (0.096), and 0.075 (0.078), respectively. Generally, the precision of SGD-SM 2.0 products performs similar with incipient products. The 245 diversities of those indicators are little between the original and reconstructed SGD-SM 2.0 products in Table 1. To a certain

extent, in-situ validation testifies the reconstructed accuracy and validity of the SGD-SM 2.0 products. These 124 selected soil moisture stations from ISMN from 2002 to 2022 are shown in Table 2, for validating SGD-SM 2.0. Besides, basic information on the representative in-situ soil moisture sites (Taking partial sites as example, including COSMOS, SD-DEM, SMOS-CBR, SCAN, PBO, USCRN and OZNET networks) is listed in Table 3. As the example, it includes the name of the station, country, longitude/latitude, main land use, lattice water, and soil organic carbon.

**Table 2.** 124 selected soil moisture stations from ISMN from 2002 to 2022 for validating SGD-SM 2.0.

| | | | | | |
|---|---|---|---|---|---|
| COSMOS-001 | COSMOS-004 | COSMOS-006 | COSMOS-007 | COSMOS-010 | COSMOS-011 |
| COSMOS-012 | COSMOS-013 | COSMOS-014 | COSMOS-015 | COSMOS-016 | COSMOS-017 |
| COSMOS-018 | COSMOS-020 | COSMOS-021 | COSMOS-023 | COSMOS-024 | COSMOS-025 |
| COSMOS-026 | COSMOS-027 | COSMOS-028 | COSMOS-029 | COSMOS-030 | COSMOS-031 |
| COSMOS-032 | COSMOS-033 | COSMOS-034 | COSMOS-035 | COSMOS-038 | COSMOS-039 |
| COSMOS-040 | COSMOS-041 | COSMOS-042 | COSMOS-043 | COSMOS-044 | COSMOS-045 |
| COSMOS-046 | COSMOS-047 | COSMOS-048 | COSMOS-049 | COSMOS-050 | COSMOS-051 |
| COSMOS-052 | COSMOS-053 | COSMOS-054 | COSMOS-055 | COSMOS-056 | COSMOS-057 |
| COSMOS-058 | COSMOS-060 | COSMOS-061 | COSMOS-062 | COSMOS-063 | COSMOS-064 |
| COSMOS-066 | COSMOS-067 | COSMOS-068 | COSMOS-069 | COSMOS-070 | COSMOS-071 |
| COSMOS-072 | COSMOS-073 | COSMOS-074 | COSMOS-076 | COSMOS-078 | COSMOS-081 |
| COSMOS-082 | COSMOS-084 | COSMOS-087 | COSMOS-089 | COSMOS-090 | COSMOS-091 |
| COSMOS-092 | COSMOS-093 | COSMOS-094 | COSMOS-095 | COSMOS-096 | COSMOS-098 |
| COSMOS-099 | COSMOS-100 | COSMOS-101 | COSMOS-102 | COSMOS-103 | COSMOS-105 |
| COSMOS-107 | COSMOS-108 | COSMOS-109 | COSMOS-110 | COSMOS-111 | COSMOS-123 |
| RSMN-15136 | RSMN-15199 | RSMN-15412 | RSMN-15470 | RSMN-15479 | SD-DEM |
| SMOS-CBR | SMOS-LHS | SMOS-MTM | SMOS-SFL | SMOS-SVN | SMOS-PZN |
| SCAN-2014 | SCAN-2046 | SCAN-2055 | SCAN-2087 | SCAN-2179 | SCAN-2181 |
| PBO-076 | PBO-094 | PBO-250 | PBO-472 | PBO-474 | PBO-482 |
| PBO-498 | PBO-508 | PBO-525 | PBO-569 | PBO-742 | PBO-811 |
| USCRN-011 | USCRN-020 | OZNET-K1 | OZNET-K2 | | |

**Table 3.** Basic information on the in-situ soil moisture sites (Taking partial sites as examples).

| Station | Lon/Lat | Elevation (m) | main land use | lattice water | soil organic carbon |
|---------|---------|---------------|---------------|---------------|---------------------|
| COSMOS-016 | 42.537, -72.171 | 316 | Crop | 4.50% | 1.59% |
| COSMOS-055 | 0.282, 36.866 | 1824 | Bush | 6.10% | 1.11% |
| COSMOS-082 | 48.141, 15.171 | 73 | Grass | 2.10% | 1.93% |
| COSMOS-096 | -14.159, 131.388 | 169 | Silty Sand | 2.30% | 1.24% |
| COSMOS-101 | -21.617, -47.632 | 563 | Grass | 1.70% | 1.87% |
| COSMOS-123 | 31.369, 91.899 | 1201 | Forest | 4.40% | 2.36% |
| SD-DEM | 13.287, 30.479 | 864 | Coarse Sand | 1.30% | 0.98% |
| SMOS-CBR | 42.563, 13.798 | 52 | Grass | 3.40% | 2.25% |
| SCAN-2014 | 38.173, -65.171 | 274 | Crop | 2.20% | 1.97% |
| PBO-076 | 24.189, -81.343 | 156 | Silty Sand | 1.90% | 1.14% |
| USCRN-011 | 20.507, -97.662 | 583 | Grass | 3.70% | 1.98% |
| OZNET-K1 | -21.683, 139.841 | 659 | Scrub | 3.60% | 2.34% |

## 4.3 Time-series validation

Long-term daily soil moisture products usually reflect typical time-series continuity (Wang et al., 2022; Seneviratne et al., 2010). Therefore, we can utilize this characteristic to validate the reliability of SGD-SM 2.0 products. As listed in Fig. 9 and Fig. 10, two time-series daily original/SGD-SM 2.0 results of 2003 to 2018, and 2005 to 2020, are given in the location (10.125°S, 42.625°W) and the location (38.375°N, 117.125°E), respectively. The blue point refers to existing valid value in Fig. 9 and Fig. 10. The red point stands for the SGD-SM 2.0 value in Fig. 9 and Fig. 10, which also represent the invalid gaps or missing soil moisture regions. The vertical coordinate denotes the percent of soil moisture product in original and SGD-SM 2.0 products. The horizontal coordinate denotes the annual date number between 2003 and 2020.

As depicted in Fig. 9(a)-(d) and Fig. 10(a)-(d), a majority of the reconstructed SGD-SM 2.0 values (in invalid gap or missing soil moisture regions) can distinctly embody the time-series continuity. In the two locations of different years, original soil moisture values and corresponding adjacent SGD-SM 2.0 values perform fore-and-aft consistency. If current valid soil moisture values behave high or low, their neighborhood SGD-SM 2.0 values also accord with them in Fig. 9 and Fig. 10. This time-series validation manifests the reliability of proposed framework and validity of our improved SGD-SM 2.0 products. Generally, the proposed SGD-SM 2.0 products are able to ensure the time-series continuity in daily temporal resolution. This point is greatly important for reconstructing long-term products. Benefiting from the utilizing of temporal information, the proposed LSTM module can extract and transmit time-series features for filling the gap and missing data regions in daily soil moisture products. Therefore, SGD-SM 2.0 can be effectively applied for global hydrology monitoring analyzing at fine

temporal scale, rather than the traditional monthly or yearly averaging operation. The former one preserves the original daily temporal resolution, while the latter one sacrifices this daily temporal resolution. This validation exactly demonstrates above significance of the proposed SGD-SM 2.0 dataset.

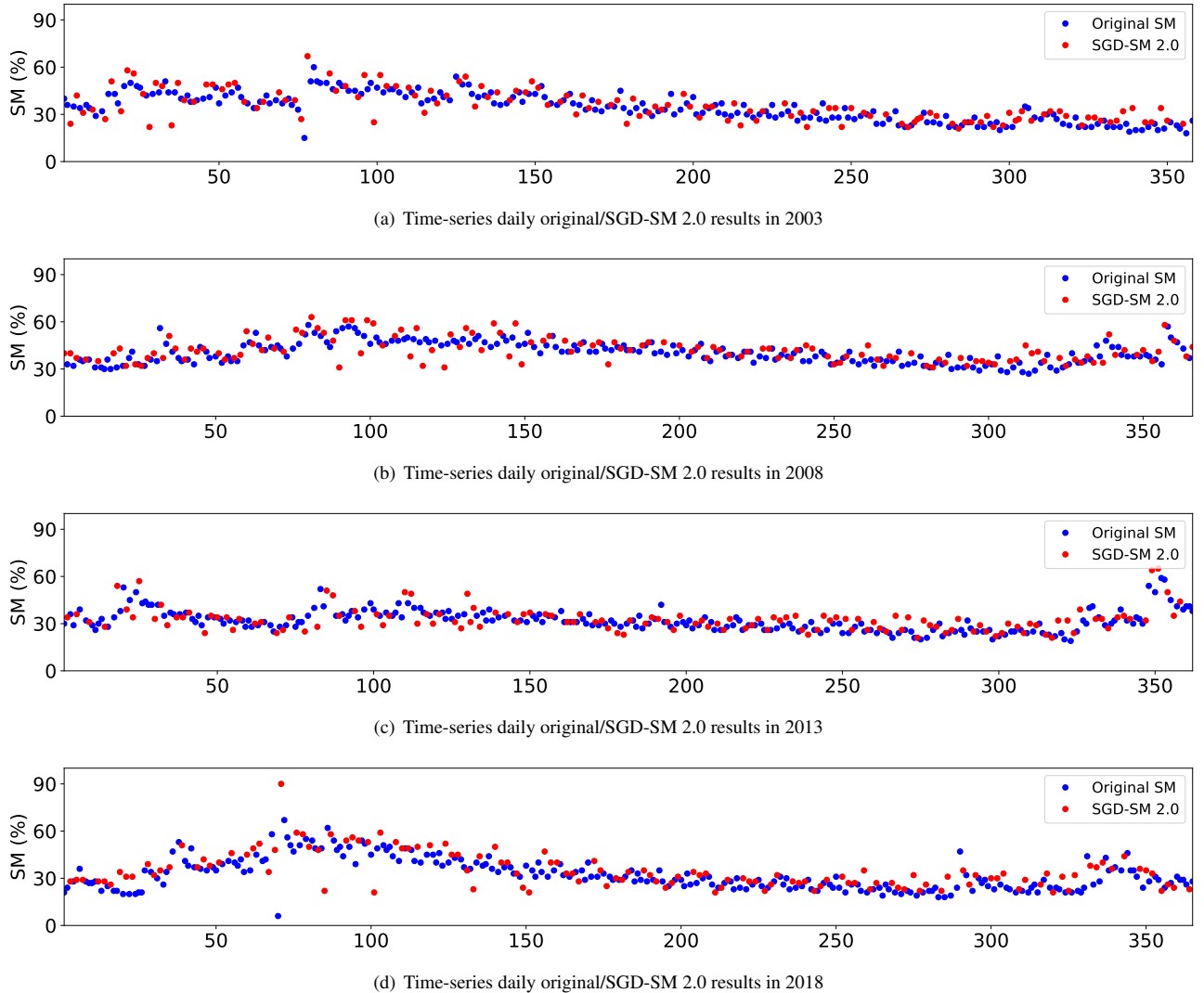

**Figure 9.** Time-series daily original/SGD-SM 2.0 results of the location (10.125°S, 42.625°W) in 2003, 2008, 2013, and 2018.

## 5 Comparisons with SGD-SM 1.0

In this section, we compare the proposed SGD-SM 2.0 dataset with previous SGD-SM 1.0 dataset, from the perspectives of reconstructing accuracy and time-series consistency. In contrast with SGD-SM 1.0, we fuse the global daily precipitation

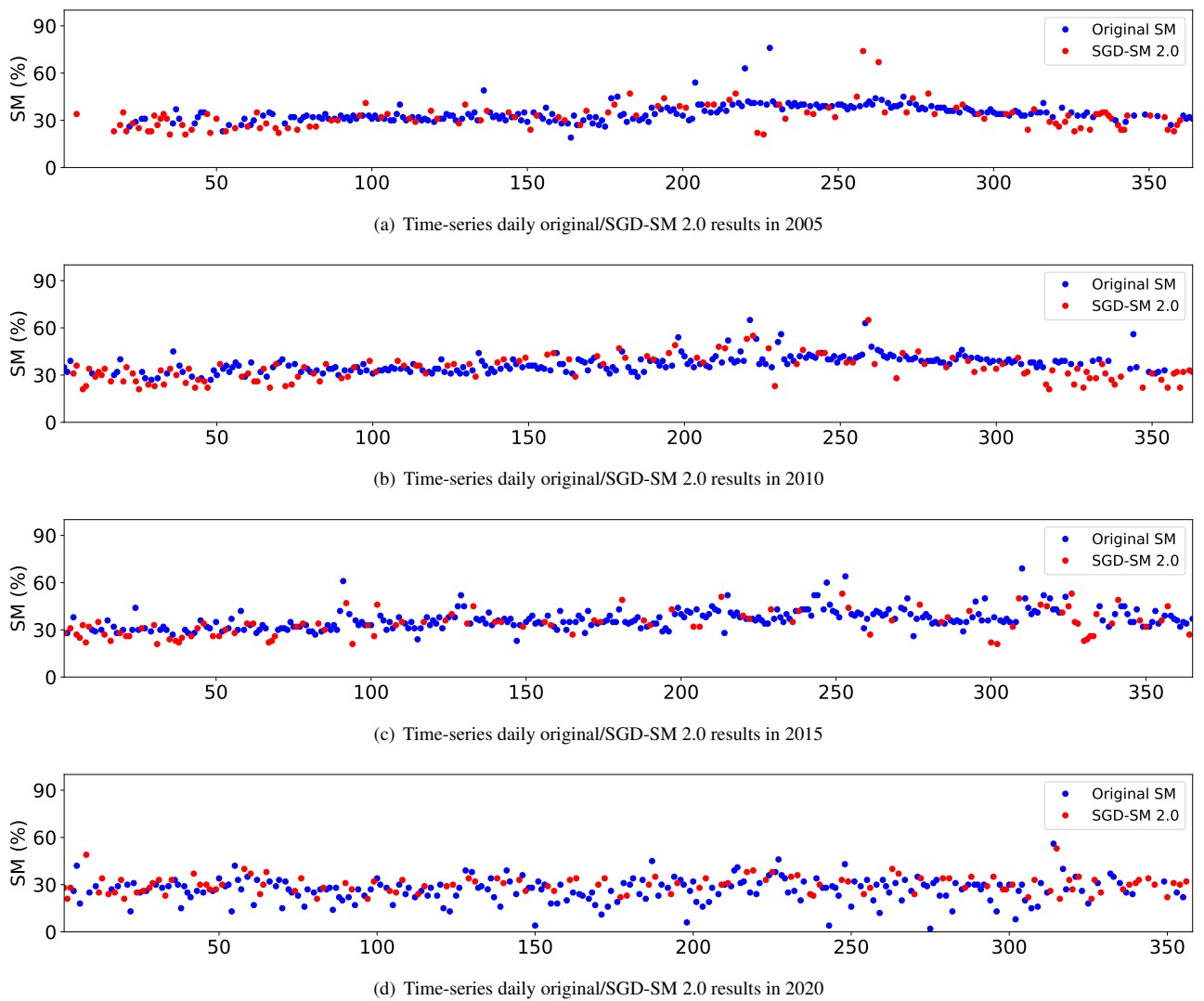

**Figure 10.** Time-series daily original/SGD-SM 2.0 results of the location (38.375°N, 117.125°E) in 2005, 2010, 2015, and 2020.

products into the reconstructing framework. In addition, the LSTM-CNN model is developed to fill the gap and missing

regions in SGD-SM 2.0 global daily soil moisture products. Detailed comparisons between the SGD-SM 1.0 and SGD-SM 2.0 are displayed as follows.

## 5.1 Reconstructing accuracy

For ensuring the same time scope with SGD-SM 1.0, we choose the part of SGD-SM 2.0 from 2013 to 2019. The average evaluation indicators (R, RMSE, and MAE) of monthly-averaging, SGD-SM 1.0 and SGD-SM 2.0 dataset by selective 124

in-situ sites are contrasted in Table 4.

Compared with monthly-averaging and SGD-SM 1.0 products, SGD-SM 2.0 products outperform on R (0.688), RMSE (0.094), and MAE (0.077). The main reason is that SGD-SM 1.0 ignores the sudden extreme weather condition for one day. If it occurs a sudden precipitation in one day, while there are no abnormalities before and after this day, SGD-SM 1.0 usually behaves with poor performance under this condition. Accordingly, SGD-SM 2.0 introduces the global daily precipitation products into the reconstructing framework. Through fusing auxiliary precipitation information, SGD-SM 2.0 products can consider the sudden extreme weather condition for single day in global daily soil moisture products. The comparisons validate the effectiveness of this point in Table 4.

**Table 4.** Comparisons between the SGD-SM 1.0 and SGD-SM 2.0 products (from 2013 to 2019) through selected 124 in-situ sites.

| Dataset version | Average evaluation indicators | | | |
| --- | --- | --- | --- | --- |
| | R | RMSE | ubRMSE | MAE |
| Monthly-Averaging | 0.612 | 0.147 | 0.089 | 0.115 |
| SGD-SM 1.0 | 0.659 | 0.107 | 0.066 | 0.083 |
| SGD-SM 2.0 | **0.688** | **0.094** | **0.058** | **0.077** |

## 5.2 Time-series consistency

Except the reconstructing accuracy, time-series consistency is also significant for seamless products (Wang et al., 2021). As portrayed in Fig. 11(a) and (b), we simultaneously depict time-series daily original soil moisture, SGD-SM 1.0/2.0, and precipitation results of the location (48.875°N, 140.375°E) in 2013, respectively. The blue point refers to existing valid values in Fig. 11. Red point stands for the SGD-SM 1.0/2.0 value in Fig. 11, which also represent the invalid gap or missing soil moisture regions. The left vertical coordinate denotes the percent of soil moisture product in original and SGD-SM 1.0/2.0 products. The right vertical coordinate refers to the daily precipitation value (unit: mm) by the IMERG precipitation products.

Compared with SGD-SM 1.0, SGD-SM 2.0 outperforms on time-series consistency in Fig. 11(a) and (b). The reconstructed SGD-SM 2.0 points behave more consecutive around their adjacent original soil moistures points than SGD-SM 1.0. While SGD-SM 1.0 exists discrete problem in Fig. 11(a), to some degree. Benefiting from the data fusion of daily precipitation information, the proposed LSTM module can extract time-series features for filling the gaps and missing regions in daily soil moisture products. Therefore, SGD-SM 2.0 can be effectively utilized for global hydrology monitoring analyzing at fine (daily) temporal resolution.

## 5.3 Uncertainty analysis

The uncertainty of SGD-SM 2.0 and proposed model could be classified as three types: 1) The errors of original AMSR-E/WindSat/AMSR2 products; 2) The meteorological factors; 3) The generalization of proposed reconstructing model.

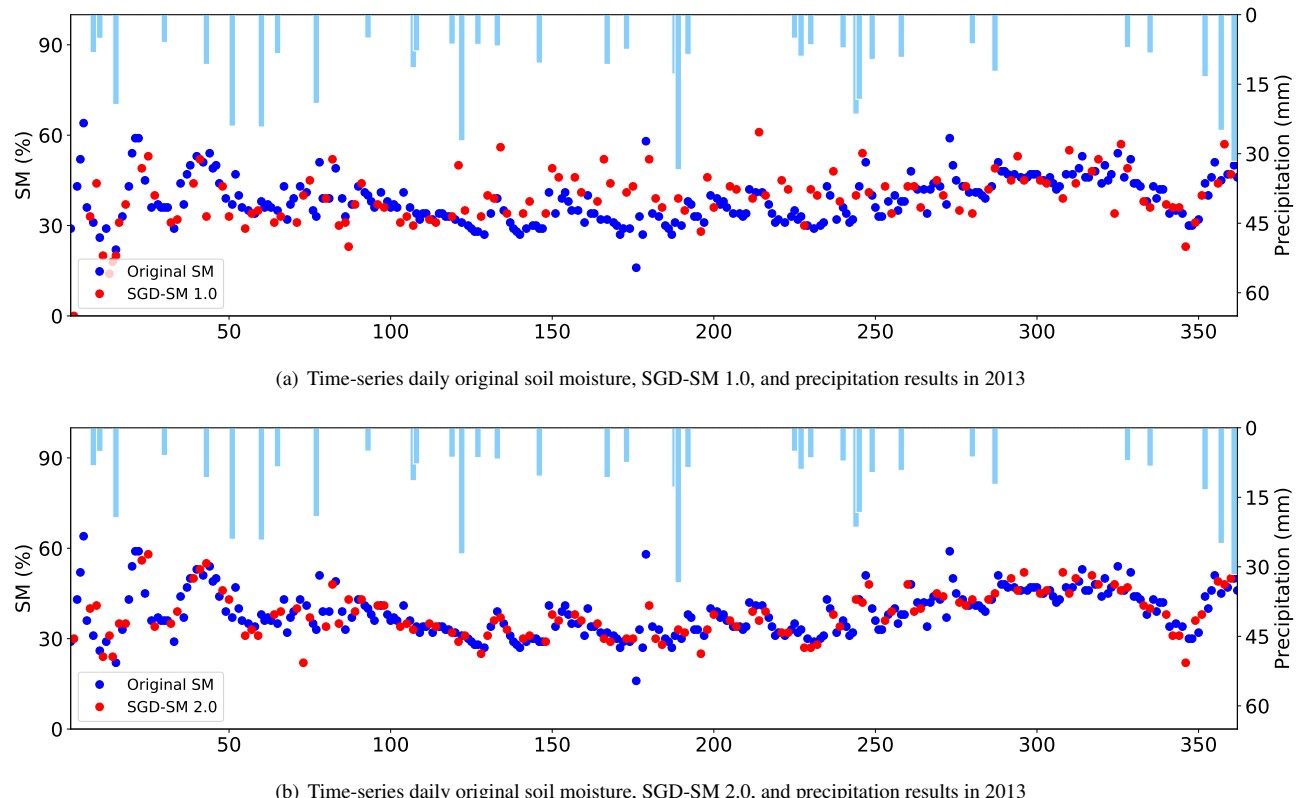

(a) Time-series daily original soil moisture, SGD-SM 1.0, and precipitation results in 2013

(b) Time-series daily original soil moisture, SGD-SM 2.0, and precipitation results in 2013

**Figure 11.** Time-series daily original soil moisture, SGD-SM 1.0/2.0, and precipitation results at location (48.875°N, 140.375°E) in 2013.

1) The errors of original AMSR-E/WindSat/AMSR2 products: The proposed SGD-SM product is generated based on original
AMSR-E/WindSat/AMSR2 products. While these passive soil moisture products also exist errors (i.e. above 0.8 m$^3$·m$^{-3}$) , due
to the satellite sensor imaging and soil moisture retrieval algorithm. As shown in Table 1, the R, RMSE, and MAE evaluation
indexes of the original products are 0.679, 0.094, and 0.075, respectively. These errors are also inevitably transmitted into the
generated SGD-SM 2.0 products. In other words, SGD-SM 2.0 absolutely trusts the initial satellite-based SM values without
any hesitation.
2) The meteorological factors: The proposed method relies on the temporal continuity and spatial consistency for daily soil
moisture gap-filling. Nevertheless, if the unusual meteorologic occurs in single day such as precipitation and snowfall, it may
disturb above assumption and influence the reconstructing effects. This uncertainty can be noticed in time-series validation,
especially for the rainy season. Although we fuse the daily precipitation products into the proposed model in SGD-SM 2.0, it
still cannot adequately reflect the emergency meteorological factors such as brief precipitation.
3) The generalization of proposed reconstructing model: In this work, we train the proposed LSTM-CNN model through
selecting complete soil moisture patches all over the world. In addition, the simulated masks are also chosen from the daily
soil moisture products. However, it still exists the differences between the training data and testing data, such as land covering

type and mask size. This uncertainty may disturb the generalization of proposed LSTM-CNN model for SGD-SM 2.0, to some degree.

## 320  6  Conclusions

In this paper, we generate an improved seamless global daily soil moisture (SGD-SM 2.0) dataset from 2002 to 2022. Compared with previous SGD-SM 1.0, the temporal range of SGD-SM 2.0 is extended to twenty years from 2002 to 2022. SGD-SM 2.0 fuses the global daily precipitation products with global daily soil moisture products. In addition, SGD-SM 2.0 develops an integrated LSTM-CNN model to fill the gaps and missing regions. In-situ validation and time-series validation

testify the soil moisture time-series of SGD-SM 2.0 products (R: 0.672, RMSE: 0.096, MAE: 0.078). In contrast with SGD-SM 1.0, the time-series curves of the improved SGD-SM 2.0 products are consistency with the original daily time-series soil moisture values.

In our future work (SGD-SM 3.0), we will fuse multi-source data such as global land cover products and land surface temperature into the reconstructed framework. More spatio-temporal model will be exploited to generate the prospective products.

In addition, we will introduce the outlier filtering strategy, to exclude these initial SM exception information (above 80 Vol.%). Identifying outliers by comparing the SM data product with the porosity information from the global soil database SoilGrid will also be utilized in SGD-SM 3.0.

*Data availability.*  The proposed SGD-SM 2.0 dataset could be acquired at **https://doi.org/10.5281/zenodo.6041561**  (Zhang et al., 2022).

*Author contributions.*  Q. Z presented the algorithm and carried out the experimental results. QQ. Y, TY. J, and MP. S polished the entire
manuscript. FJ. S downloaded soil moisture products and precipitation products in this work. All authors checked and gave related comments for this work.

*Competing interests.*  The authors declare that they have no conflict of interest.

*Acknowledgements.*  This work was supported in part by the National Natural Science Foundation of China under Grant 41922008, 61971319 and 61971082. We appreciatively acknowledge GES DISC and ISMN, for them releasing related products and in-situ sites.

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
