# Peer review of "SGD-SM 2.0: An Improved Seamless Global Daily Soil Moisture Long-term Dataset From 2002 to 2022"

_Earth System Science Data, 2022_

## Author Comment (AC1)

**Response to the Comments of Referee #1**

Dear Referee #1:

We are particularly grateful for your careful reading, and for giving us many constructive comments of this work!

According to the comments and suggestions, we have tried our best to improve the previous manuscript ESSD-2022-80 (SGD-SM 2.0: An Improved Seamless Global Daily Soil Moisture Long-term Dataset From 2002 to 2022). An item-by-item response follows.

Once again, we are particularly grateful for your careful reading and constructive comments. Thanks very much for your time.

Best regards,

Qiang Zhang

**General comments:**

> *This paper develops SGD-SM 2.0 framework for reconstruction of seamless global daily soil moisture dataset from 2002 to 2022, based on the development of the LSTM-CNN method. The method fuses the global daily precipitation products and is able to consider sudden extreme weather condition. Generally, the topic is interesting, the method makes sense and the results are supportive. Some minor comments before positive publication are as follows.*

**Response:** We are particularly grateful to the reviewer for his/her detailed suggestions! According to the comments, we have tried our best to improve the previous manuscript. An item-by-item response to each constructive comment follows.

**Major comments:**

> **Q1.1:** *Please provide the parameters and descriptions of AMSR-E, AMSR2, and WindSat to demonstrate the rationality of using three heterogeneous sensors.*

**Response:** Thanks for this suggestion. The parameters and descriptions of AMSR-E, AMSR2, and WindSat are provided below:

AMSR-E/2 and WindSat global daily soil moisture products are utilized from 2002 to 2022. These three sensors are onboarded at Aqua satellite, GCOM-W1 and Coriolis satellite, respectively (Nepal et al., 2021). AMSR-E, AMSR2 and WindSat are all passive sensors for soil moisture retrieving. The spatial resolution is all 0.25° grid (about 25km) in these products, as depicted in Fig. 1(a)-(c). The retrieving model adopts the land parameter retrieval model (LPRM) for AMSR-E, WindSat, and AMSR2 products (McColl et al., 2017). We select the descending orbit (night-time), and 6.9 GHz band for all these soil moisture products.

**Q1.2:** *The original SM data acquired from AMSR-E, AMSR2, and WindSat sensors were used to generate the seamless soil moisture dataset. As we know, although the frequency band of both AMSR-E/2 and WindSat have the same frequency band to retrieve the soil moisture, the GHz of WindSat sensor is different from that of AMSR-E/2. Is there a big difference in accuracy between 2011.10.5 to 2012.07.02 (i.e., using WindSat) and other periods (i.e., using AMSR-E/2)?*

**Response:** Thanks for this meaningful question. As the reviewer stated, although the frequency band of both AMSR-E/2 and WindSat have the same frequency band to retrieve the soil moisture, the GHz of WindSat sensor is different from that of AMSR-E/2. Between 2011.10.5 to 2012.07.02, both the original SM data acquired from AMSR-E and WindSat is existing. The main difference between WindSat and AMSR-E is the global daily land coverage rate. The average land coverage rate of WindSat is just 34%, compared with AMSR-E (about 50%). In other word, AMSR-E outperforms on global reconstruction accuracy than WindSat on SGD-SM 2.0, due to the quantity of valid information.

**Q1.3:** *Section 4.3 only exhibits the dynamic change of the SGD-SM 2.0 dataset. Perhaps, the advantages of version 2.0 can be demonstrated by introducing version 1.0 as the reference in this section. Moreover, this revised description is different from Section 5.2 (the time series in the precipitation area).*

**Response:** Thanks for this issue. In Section 4.3, we exhibit both the dynamic change of original and reconstructed soil moisture (SGD-SM 2.0) value. The main purpose is to reveal the typical time-series continuity in this time-series validation. In Section 5.2, we provide the time-series comparisons between SGD-SM 1.0 and 2.0 products with time-series precipitation data. The main purpose is to compare the difference of SGD-SM 1.0 and 2.0, especially for the sudden daily

precipitation weather. Though this comparison in Section 5.2, the advantage of SGD-SM 2.0 could be better reflected via daily precipitation data assimilation and LSTM-CNN model. Therefore, we arrange two different time-series validation sections to demonstrate the advantage and availability of SGD-SM 2.0.

**Q1.4:** *Please provide the website for collection of in-situ data, if the data are public.*

**Response:** Thanks for this comment. We have supplemented the website for collection of in-situ data in the revised manuscript. These in-situ soil moisture data are public and could be downloaded at https://ismn.geo.tuwien.ac.at/en/.

**Q1.5:** *Why the partial CNN and mask updating were used to reconstruct the missing regions?*

**Response:** Thanks for mentioning this query. For partial CNN, it can effectively acquire the spatial information within valid regions, and eliminate the invalid information within gap or soil moisture missing regions. For mask updating, if the partial convolution can generate at least one valid value of the output result, we need mark this location as valid value in the new masks. We have supplemented these explanations in Section 3.1.

**Q1.6:** *What is the relation between the two sub figures in Fig. 1?*

**Response:** Thanks for this question. In Fig. 1, two sub captions of original soil moisture products of AMSR-E and WindSat are incorrect. We have revised this mistake in current version.

[Figure]

(a) Original SM products of AMSR-E in 2009.6.1    (b) Original SM products of WindSat in 2012.1.9

**Fig. 1.** Daily soil moisture products of AMSR-E and WindSat.

**Q1.7:** *Please consider including more up-to-date references on gap filling, such as [Remote sensing image gap filling based on spatial-spectral random forests. Science of Remote Sensing, 2022, 5: 100048].*

**Response:** Thanks for this comment. We have cited this reference [Remote sensing image gap filling based on spatial-spectral random forests. Science of Remote Sensing, 2022, 5: 100048] in the revised manuscript.

**Q1.8:** *Check the caption of Fig. 6.*

**Response:** Thanks for this suggestion. In Fig. 6, the "Original SM and proposed SGD-SM 2.0 results in 10, 20, and 30 September 2002.4" has be recorrected as "Original SM and proposed SGD-SM 2.0 results in 10, 20, and 30 September 2002".

---

## Author Comment (AC2)

**Response to the Comments of Referee #2**

Dear Referee #2:

We are particularly grateful for your careful reading, and for giving us many constructive comments of this work!

According to the comments and suggestions, we have tried our best to improve the previous manuscript ESSD-2022-80 (SGD-SM 2.0: An Improved Seamless Global Daily Soil Moisture Long-term Dataset From 2002 to 2022). An item-by-item response follows.

Once again, we are particularly grateful for your careful reading and constructive comments. Thanks very much for your time.

Best regards,

Qiang Zhang

**General comments:**

> *This manuscript presents a novel study on use of three sensors to reconstruct SGD-SM 2.0 products. One of the novel aspects of this study is that global daily precipitation products are wisely assimilated into the proposed LSTM-CNN, to fill gaps in daily soil moisture products. This methodology represents a substantial advancement in generating global soil moisture products that synergistically incorporate soil moisture and its closely associated hydrological variable, precipitation from the last precipitation satellite. The improved SGD-SM 2.0 product has been shown to outperform the previous SGD-SM 1.0 product in terms of accuracy and time-series consistency. I recommend accepting this wonderful work after minor revision.*

**Response:** We are particularly grateful to the reviewer for his/her detailed suggestions! According to the comments, we have tried our best to improve the previous manuscript. An item-by-item response to each constructive comment follows.

**Major comments:**

**Q2.1:** *Page 2 Line 24: AMSR2 and WindSat products in caption (a) and (b) are incorrect.*

**Response:** Thanks for this comment. In Fig. 1, two sub captions of original soil moisture products of AMSR-E and WindSat are incorrect. We have revised this mistake in current version.

**Q2.2:** *Page 3 Line 64: Word 'description' is repetitive in this sentence.*

**Response:** Thanks for reviewer's careful checking. We have revised this sentence as "Sect. 2 provides a description of products and data used in this work" in current version.

**Q2.3:** *Page 5 Line 101: IMERG precipitation products should be given the full name.*

**Response:** Thanks for this suggestion. We have given the full name of IMERG (Integrated Multi-satellitE Retrievals for GPM) in Section 2.2.

**Q2.4:** *Page 9 Line 185: Why did the authors use the global land mask $M_L$ in the loss function?*

**Response:** Thanks for this meaningful query. The global soil moisture uniformity and local soil moisture heterogeneity are both taken into consideration in the proposed LSTM-CNN reconstructing model. Therefore, we use the global land mask $M_L$ in the loss function to better reconstruct the gap regions. We have supplemented these explanations in Section 3.1.

**Q2.5:** *Page 12 Line 215: 'drawing into global daily precipitation products' should be revised as 'fusing global daily precipitation products.'*

**Response:** Thanks for mentioning this issue. We have revised this sentence as "Through fusing global daily precipitation products, SGD-SM 2.0 can consider the sporadic extreme weather condition for single day." in Section 4.1.

**Q2.6:** *Page 16 Line 281: The authors claimed that the reconstructed SGD-SM 2.0 points behave more consecutive around their adjacent original soil moistures points than SGD-SM 1.0. More explanations need to be given for this attribute.*

**Response:** Thanks for this suggestion. Compared with SGD-SM 1.0, SGD-SM 2.0 outperforms on time-series consistency in Fig. 11(a) and (b). The reconstructed SGD-SM 2.0 points behave

more consecutive around their adjacent original soil moistures points than SGD-SM 1.0. While SGD-SM 1.0 exists discrete problem in Fig. 11(a), to some degree. Benefiting from the data fusion of daily precipitation information, the proposed LSTM module can extract time-series features for filling the gaps and missing regions in daily soil moisture products. Therefore, SGD-SM 2.0 can be effectively utilized for global hydrology monitoring analyzing at fine temporal scale, rather than the traditional monthly or yearly averaging operation.

[Figure]

(a) Time-series daily original soil moisture, SGD-SM 1.0, and precipitation results in 2013

(b) Time-series daily original soil moisture, SGD-SM 2.0, and precipitation results in 2013

**Fig. 11.** Time-series daily original soil moisture, SGD-SM 1.0/2.0, and precipitation results at location (48.875°N, 140.375°E) in 2013.

**Q2.7:** *Page 17 Line 297: Data availability. Current descriptions about SGD-SM 2.0 in the website are not Data availability. The authors may want to supplement specific information for possible users.*

**Response:** Thanks for this comment. We have supplemented more specific information for SGD-

SM 2.0 at https://doi.org/10.5281/zenodo.6041561.

[Figure]

Current descriptions about SGD-SM 2.0 at the websitehttps://doi.org/10.5281/zenodo.6041561.

---

## Author Comment (AC3)

**Response to the Comments of Referee #3**

Dear Referee #3:

We are particularly grateful for your careful reading, and for giving us many constructive comments of this work!

According to the comments and suggestions, we have tried our best to improve the previous manuscript ESSD-2022-80 (SGD-SM 2.0: An Improved Seamless Global Daily Soil Moisture Long-term Dataset From 2002 to 2022). An item-by-item response follows.

Once again, we are particularly grateful for your careful reading and constructive comments. Thanks very much for your time.

Best regards,

Qiang Zhang

**General comments:**

*This paper presents an improved seamless global daily soil moisture dataset from 2002 to 2022 (SGD-SM 2.0) based on the three satellite soil moisture sensors AMSR-E, AMSR2 and WindSat and Global daily precipitation products. A new convolutional neural network approach is used to fill the gaps and missing regions and ISMN data is used for the validation.*

*The topic is of general interest due to the increasing drought and overexploitation of water resources in many regions of the world due to global climate change and fits well within the scope of the ESSD. The applied methods mostly appropriate and the manuscript is mostly well written but contains some incorrect wording and phrasing (see specific comments). My main concern is that the authors used only six stations for the validation of the global SGD-SM 2.0 data set, which is not inappropriate. The authors should make an effort to test whether SGD-SM 2.0 data accuracy is independent on the environmental conditions. The SGD-SM 2.0 data product would be well received by the science community working on Global Change issues and can be recommended for publication after all issues detailed below have been appropriately addressed.*

**Response:** We are particularly grateful to the reviewer for his/her detailed suggestions! According to the comments, we have tried our best to improve the previous manuscript. For the in-situ validation issue, more specific explanations could be checked in Q3.1. For the environmental condition issue, the discussion on whether SGD-SM 2.0 data accuracy is independent could also be checked in Q3.1. An item-by-item response to each constructive comment follows.

**Major comments:**

**Q3.1:** *The authors only show the averaged evaluations indicators from all selected ISMN stations and only six in-situ soil moisture stations were actually used for the validation of global SGD-SM*

*2.0 data set. In my view, this is not enough to appropriately demonstrate the accuracy of a global SM data set. There much more data is available at ISMN. In addition, other in-situ soil moisture data products are freely available, e.g. Bogena et al. (2022). In this way, potential users could also see if the SGD-SM 2.0 data accuracy is independent on the environmental conditions, e.g. soil properties, vegetation coverages, climate zone.*

**Response:** Thanks for this comment. In this work, we select 124 stations from ISMN from 2002 to 2022 and match them with corresponding soil moisture product in SGD-SM 2.0. Actually, we chose six in-situ soil moisture stations as examples for scatter visualization. In other words, all the selected 124 in-situ sites are employed to validate the accuracy of SGD-SM 2.0. We match the hourly in-site values with the descending products. In consideration of validation reliability, we choose the two neighboring in-site values correspond with the observation time of soil moisture products. Then we average them as the ground-truth data.

Through all the 124 selected in-situ sites, Table 1 compares the original products with SGD-SM 2.0. The average evaluation indicators (R, RMSE, and MAE) of original soil moisture and SGD-SM 2.0 products are 0.679 (0.672), 0.094 (0.096), and 0.075 (0.078), respectively. Generally, the precision of SGD-SM 2.0 products performs similar with incipient products. The diversities of those indicators are little between the original and reconstructed SGD-SM 2.0 products in Table 1. To a certain extent, in-situ validation testifies the reconstructed accuracy and validity of the SGD-SM 2.0 products.

**Table 1.** Comparisons between the original and SGD-SM 2.0 products through 124 selected in-situ sites.

| Soil moisture products | Average evaluation indicators | | | |
|:---:|:---:|:---:|:---:|:---:|
| | R | RMSE | ubRMSE | MAE |
| Original | 0.679 | 0.094 | 0.058 | 0.075 |
| SGD-SM 2.0 | 0.672 | 0.096 | 0.061 | 0.078 |

In terms of the independent on the environmental conditions (e.g. soil properties, vegetation coverages, climate zone), these 124 selected in-situ sites are widely distributed all over the world

(Europe, North America, South America, Asian, Africa and Australia). The soil properties, vegetation coverages and climate zones are diverse from each other. Through this in-situ validation way, we can test whether SGD-SM 2.0 data accuracy is independent on the environmental conditions. These descriptions have been supplemented in the revised manuscript.

**Q3.2:** *Some soil moisture data shown in Fig. 8 show extremely SM high values of more than 80 Vol.%. Such high values are very unlikely, as soil porosity in most soil is typically between 40-50 Vol.%, indicating measurement errors in the in-situ data or soils with extremely high organic matter or clay content. Indicating a reference site description will help to understand this better. On the other hand, the SGD-SM 2.0 data the same high values, which is astonishing. In my view, these data outliers could be the result of SM overestimation by the CNN procedure due to the precipitation consideration. In addition, single outliers can be found in Figs. 9d and 10a. Again, this indicates the influence of precipitation. Maybe the data should be cleaned with an outlier detection method? Please add at least a discussion on these issues.*

**Response:** Thanks for these issues. Actually, the SM values in this work are the volume ratio (unit: $m^3 \cdot m^{-3}$, from 0% to 100%), rather than the mass ratio ($kg \cdot m^{-3}$, usually 0% to 50%). This phenomenon is normal because of the unit via volume ratio, not measurement errors or SM overestimation by the CNN. For the outliers in Figs. 9d and 10a, this indeed indicates the influence of precipitation for the proposed LSTM-CNN model. We also consider the outlier detection method, while filter strategy will also disturb the maximal/minimum value. Overall, these outliers are few with small impact for SGD-SM 2.0. Therefore, we don't clean the data with an outlier detection method. The future work on SGD-SM 3.0 will develop a new framework to restrain the outlier problem. These descriptions have been supplemented in the revised manuscript.

**Q3.3:** *The in-situ soil moisture data from ISMN are treated anonymously in this work. However, the site owners that work hard to maintain the soil moisture stations should be better cited. This will help the site owners to ensure funds for the costly operation of the stations and data management. Therefore, the authors should add at table with basic information on the soil moisture data using, including the name of the site owners and/or monitoring networks instead of just presenting the station coordinates. See Bogena et al. (2022) for a great example. The necessary information is available in the metadata descriptions at ISMN.*

*Literature: Bogena, H.R., M. Schrön, J. Jakobi, P. Ney, S. Zacharias, M. Andreasen, R. Baatz, . . . and H. Vereecken (2022): COSMOS-Europe: A European network of Cosmic-Ray Neutron Soil Moisture Sensors. Earth Syst. Sci. Data 14: 1125–1151. DOI: 10.5194/essd-14-1125-2022*

**Response:** Thanks for this significant suggestion. We have added a table with basic information on the in-situ soil moisture sites like Bogena et al. (2022). As listed in Table 2, it includes the name of the station, country, longitude/latitude, main land use, lattice water, and soil organic carbon. Due to the page limiting, we give the six COSMOS in-situ sites in Fig. 8 as follow:

**Table 2.** Basic information on the six COSMOS in-situ soil moisture sites in Fig. 8.

| Station | Lon/Lat | Elevation (m) | main land use | lattice water | soil organic carbon |
|---------|---------|---------------|---------------|---------------|---------------------|
| COSMOS-016 | 42.537, -72.171 | 316 | Crop | 4.50% | 1.59% |
| COSMOS-055 | 0.2825, 36.866 | 1824 | Bush | 6.10% | 1.11% |
| COSMOS-082 | 48.141, 15.171 | 73 | Grass | 2.10% | 1.93% |
| COSMOS-096 | -14.159, 131.388 | 169 | Silty Sand | 2.30% | 1.24% |
| COSMOS-101 | -21.617, -47.632 | 563 | Grass | 1.70% | 1.87% |
| COSMOS-123 | 31.369, 91.899 | 1201 | Forest | 4.48% | 2.36% |

[Literature: Bogena, H. R., Schrön, M., Jakobi, J. et al.: COSMOS-Europe: a European network of cosmic-ray neutron soil moisture sensors, Earth Syst. Sci. Data, 14, 1125–1151, https://doi.org/10.5194/essd-14-1125-2022, 2022.]

**Q3.4:** *Throughout the manuscript, you use the term "assimilation" in the context of including precipitation data in your CNN based data interpolation method. However, I think this is not appropriate as the term "data assimilation" is generally used optimally combine numerical models with observations.*

**Response:** Thanks for this issue. We also agree that "data assimilation" is generally used optimally combine numerical models with observations. In this work, SGD-SM 2.0 introduces the global daily precipitation products into the reconstructing framework. Through the auxiliary precipitation data, SGD-SM 2.0 could lead in the daily extreme weather information for gap-filling. Therefore, we have replaced "assimilation" as "fusion" in the whole manuscript, to better embody the meaning of multi-source products fusion (precipitation and soil moisture).

**Specific comments:**

**Q3.5:** *L17: Please cite the more recent ISMN publication of Dorigo et al. (2021).*
*Literature: Dorigo, W., I. Himmelbauer, D. Aberer, L. Schremmer, I. Petrakovic, L. Zappa, W. Preimesberger, A. Xaver, F. Annor, J. Ardö, D. Baldocchi, M. Bitelli, G. Blöschl, H. Bogena, . . . and R. Sabia (2021): The International Soil Moisture Network: serving Earth system science for over a decade. Hydrol. Earth Syst. Sci. 25: 5749–5804. DOI:10.5194/hess-25-5749-2021*

**Response:** Thanks for this comment. We have cited this publication in the revised manuscript as follow:

[Citation: Dorigo, W., Himmelbauer, I., Aberer, D. et al.: The International Soil Moisture Network: serving Earth system science for over a decade, Hydrol. Earth Syst. Sci., 25, 5749–5804, https://doi.org/10.5194/hess-25-5749-2021, 2021.]

**Q3.6:** *L21-22: Incorrect phrasing.*

**Response:** Thanks for this comment. We have revised this sentence as follow:

"As shown in Fig. 1(a) and (b), these soil moisture products exist plenty of gap regions."

**Q3.7:** *L23: Change to "approximately 20% to 80%".*

**Response:** Thanks for this suggestion. We have rewritten this sentence as follow:

"Actually, the land coverage rate is only approximately 20% to 80% in daily AMSR-E/2 and WindSat quantitative products."

**Q3.8:** *L30: "words".*

**Response:** Many thanks for the reviewer's careful reading and checking! We have revised "word" as "words" in this sentence.

**Q3.9:** *L31: Change "destroys" to "degrades" or similar.*

**Response:** Thanks for pointing out this issue. We have changed "destroys" to "degrades" in this sentence.

**Q3.10:** *L35-36: Citation is missing.*

**Response:** Thanks for this comment. We have supplemented the related citation in the revised manuscript as follow:

"Relevant quantitative indexes (R, RMSE and MAE) and results demonstrate that SGD-SM 1.0 products can be extended for global, daily and full-coverage soil moisture measurements (Zhang et al., 2021)."

[Citation: Zhang, Q., Yuan, Q., Li, J., Wang, Y., Sun, F., and Zhang, L.: Generating seamless global daily AMSR2 soil moisture (SGD-SM) long-term products for the years 2013–2019, Earth Syst. Sci. Data, 13, 1385–1401, https://doi.org/10.5194/essd-13-1385-2021, 2021.]

**Q3.11:** *L43-44: Reads awful, please rewrite.*

**Response:** Thanks for this suggestion. We have rewritten this sentence as follow:

"SGD-SM 1.0 ignores the daily extreme weather condition. If one day occurs a sudden precipitation, SGD-SM 1.0 usually performs poor under this scenario."

**Q3.12:** *L56-57: Incorrect phrasing.*

**Response:** Thanks for this comment. We have rewritten this incorrect phrasing in this sentence as follow:

"Through fusing auxiliary precipitation data, SGD-SM 2.0 could lead in the daily extreme weather information for gap-filling."

**Q3.13:** *L70: Please mention the source of the in-situ data.*

**Response:** Thanks for this suggestion. We have given the detailed source of the in-situ data as follow:

"The in-situ soil moisture sites are employed to validate the reconstructing precision of SGD-SM 2.0. These in-situ data are downloaded from International Soil Moisture Network (ISMN)."

**Q3.14:** *L78: The GES DISC website should be referenced.*

**Response:** Thanks for this comment. We have referenced The GES DISC website in this sentence:

"These datasets are all recorded at GES DISC website (NASA GES DISC, 2022)."

[Reference: NASA GES DISC: https://disc.gsfc.nasa.gov/, last access: 06 June 2022.]

**Q3.15:** *L85: Reads awful, please rewrite.*

**Response:** Thanks for this suggestion. We have rewritten this sentence as follow:

"Precipitation usually has a high correlation with soil moisture in the corresponding regions."

**Q3.16:** *L97: Please cite the more recent ISMN publication of Dorigo et al. (2021).*

**Response:** Thanks for this comment. We have cited this publication in the revised manuscript as follow:

[Citation: Dorigo, W., Himmelbauer, I., Aberer, D. et al.: The International Soil Moisture Network: serving Earth system science for over a decade, Hydrol. Earth Syst. Sci., 25, 5749–5804, https://doi.org/10.5194/hess-25-5749-2021, 2021.]

**Q3.17:** *L103: Change here and elsewhere to "long and short-term".*

**Response:** Thanks for this issue. We have revised this statement as "long and short-term memory" in this sentence and elsewhere of the updated version.

**Q3.18:** *L130: Change to "soil moisture and precipitation products".*

**Response:** Thanks for pointing out this error. We have recorrected this sentence as "soil moisture and precipitation products" in the revised manuscript.

**Q3.19:** *L132-133: Can you estimate the average time scales of the long and short-term memories and their variabilities? It would be interesting to know how different the time scales are.*

**Response:** Thanks for this interesting query. The proposed model uses long and short-term memory network to extract time-series information for generating SGD-SM 2.0. Actually, this network cannot estimate the average time scales of the long and short-term memories and their variabilities. The memory mechanism introduces the short-term memory to ensure the adjacent correction for the next node. The long-term memory is used to ensure the sequentiality of time-series nodes.

**Q3.20:** *L190: The term "epoch number" should be explained.*

**Response:** Thanks for this issue. We have explained the definition of the term "epoch number" in the updated version as follow: "One epoch represents that all the samples in the training set have been utilized for the neural network optimization at one time."

**Q3.21:** *L290: Change to "the soil moisture time-series of".*

**Response:** Thanks for this comment. We have revised this sentence as "In-situ validation and time-series validation testify the soil moisture time-series of SGD-SM 2.0 products (R: 0.672, RMSE: 0.096, MAE: 0.078)" in Section 6.

**Q3.22:** *Figure 11: Please show the precipitation in reverse order and as bar chart, which is the standard way of presenting precipitation and much better to understand.*

**Response:** Thanks for this meaningful suggestion. We have shown the precipitation in reverse order and as bar chart in Fig. 11. Current figure is much better to understand the significance of precipitation information.

[Figure]

(a) Time-series daily original soil moisture, SGD-SM 1.0, and precipitation results in 2013

[Figure]

(b) Time-series daily original soil moisture, SGD-SM 2.0, and precipitation results in 2013

**Fig. 11.** Time-series daily original soil moisture, SGD-SM 1.0/2.0, and precipitation results at location (48.875°N, 140.375°E) in 2013.

---

## Author Comment (AC4)

**Response to the Comments of Referee #4**

Dear Referee #4:

We are particularly grateful for your careful reading, and for giving us many constructive comments of this work!

According to the comments and suggestions, we have tried our best to improve the previous manuscript ESSD-2022-80 (SGD-SM 2.0: An Improved Seamless Global Daily Soil Moisture Long-term Dataset From 2002 to 2022). An item-by-item response follows.

Once again, we are particularly grateful for your careful reading and constructive comments. Thanks very much for your time.

Best regards,

Qiang Zhang

**General comments:**

*This paper addresses the emergent need to increase soil moisture information access, quality, and quantity for multiple users and applications. The authors present an interesting study about reporting a new data version of a seamless global soil moisture product that increases both the quality and accuracy of the previous version of this product. The methods are sound and novel, particularly the development of a deep learning algorithm to fill daily gaps in soil moisture estimates. The authors compare the old and new versions of the datasets, and they provide a robust quantitative accuracy benchmark between versions.*

**Response:** We are particularly grateful to the reviewer for his/her detailed suggestions! According to the comments, we have tried our best to improve the previous manuscript. An item-by-item response to each constructive comment follows.

**Major comments:**

**Q4.1:** *The paper is generally well written. However, from the narrative, I feel that there are missing technical details. For example, the use and role of the three passive microwave sensors in modeled soil moisture values in the presence of the precipitation dataset is unclear. Also, can the authors elaborate on prediction variance or model-based uncertainties? I feel that uncertainty of estimates is commonly not presented in soil moisture gap-filling efforts despite being helpful for assessing the reliability of soil moisture predictions.*

**Response:** Thanks for this comment. For the use and role of three passive microwave sensors (AMSR-E, AMSR2 and WindSat) in the presence of the precipitation dataset, we have supplemented more detailed expatiations in our revised manuscript:

[revised manuscript text omitted]

**4.2:** *It is also my opinion that the accuracy limitations or advantages of the new product version are relative to the reader. For example, the authors poorly discuss their accuracy findings against previous research or gap filling efforts of satellite soil moisture estimates across scales.*

**Response:** Thanks for this issue. The accuracy findings against previous research have been supplemented in current manuscript. We compare the proposed SGD-SM 2.0 dataset with previous SGD-SM 1.0 dataset, from the perspectives of reconstructing accuracy and time-series consistency. In contrast with SGD-SM 1.0, we fuse the global daily precipitation products into the reconstructing framework. In addition, the LSTM-CNN model is developed to fill the gap and missing regions in SGD-SM 2.0 global daily soil moisture products.

Compared with SGD-SM 1.0 products, SGD-SM 2.0 products outperform on R (0.688), RMSE (0.094), and MAE (0.077). The main reason is that SGD-SM 1.0 ignores the sudden extreme weather condition for one day. If it occurs a sudden precipitation in one day, while there are no abnormalities before and after this day, SGD-SM 1.0 usually behaves with poor performance under this condition. Accordingly, SGD-SM 2.0 introduces the global daily precipitation products into the reconstructing framework. Through fusing auxiliary precipitation information, SGD-SM 2.0 products can consider the sudden extreme weather condition for single day in global daily soil moisture products. The comparisons validate the effectiveness of this point in Table 3.

**Table 3.** Comparisons between the SGD-SM 1.0 and SGD-SM 2.0 products (from 2013 to 2019) through selected 124 in-situ sites.

| Dataset version | Average evaluation indicators | | | |
|:---:|:---:|:---:|:---:|:---:|
| | R | RMSE | ubRMSE | MAE |
| Monthly-Averaging | 0.612 | 0.147 | 0.089 | 0.115 |
| SGD-SM 1.0 | 0.659 | 0.107 | 0.066 | 0.083 |
| SGD-SM 2.0 | **0.688** | **0.094** | **0.058** | **0.077** |

Except the reconstructing accuracy, time-series consistency is also significant for generating

seamless daily products (Wang et al., 2021). As portrayed in Fig. 11(a) and (b), we simultaneously depict time-series daily original soil moisture, SGD-SM 1.0/2.0, and precipitation results of the location (48.875°N, 140.375°E) in 2013, respectively. The blue point refers to existing valid values in Fig. 11. Red point stands for the SGD-SM 1.0/2.0 value in Fig. 11, which also represent the invalid gap or missing soil moisture regions. The left vertical coordinate denotes the percent of soil moisture product in original and SGD-SM 1.0/2.0 products. The right vertical coordinate refers to the daily precipitation value (unit: mm) by the IMERG level 3 global daily final precipitation products. The horizontal coordinate denotes the date number in 2013.

[Figure]

(a) Time-series daily original soil moisture, SGD-SM 1.0, and precipitation results in 2013

(b) Time-series daily original soil moisture, SGD-SM 2.0, and precipitation results in 2013

**Fig. 11.** Time-series daily original soil moisture, SGD-SM 1.0/2.0, and precipitation results at location (48.875°N, 140.375°E) in 2013.

Compared with SGD-SM 1.0, SGD-SM 2.0 outperforms on time-series consistency in Fig. 11(a) and (b). The reconstructed SGD-SM 2.0 points behave more consecutive around their adjacent original soil moistures points than SGD-SM 1.0. While SGD-SM 1.0 exists discrete problem

in Fig. 11(a), to some degree. Benefiting from the data fusion of daily precipitation information, the proposed LSTM module can extract time-series features for filling the gaps and missing regions in daily soil moisture products. Therefore, SGD-SM 2.0 can be effectively utilized for global hydrology monitoring analyzing at fine temporal scale, rather than the traditional monthly or yearly averaging operation.

**Q4.3:** *The first version of the product has a relatively good number of citations, meaning that the community uses the product and that the methodological approach is being compared with similar research. The authors provide a thorough comparison between product versions, but they do not present the discussion of findings against previous research. I would appreciate more discussion about the potential implications of using the product's old or new version in multiple applications in terms of other available soil moisture estimates.*

**Response:** Thanks for this meaningful suggestion. We have provided more discussions about the potential implications of using the product's old or new version. Compared with monthly-averaging and SGD-SM 1.0 products, SGD-SM 2.0 products outperform on R (0.688), RMSE (0.094), and MAE (0.077). The main reason is that SGD-SM 1.0 ignores the sudden extreme weather condition for one day. If it occurs a sudden precipitation in one day, while there are no abnormalities before and after this day, SGD-SM 1.0 usually behaves with poor performance under this condition. Accordingly, SGD-SM 2.0 introduces the global daily precipitation products into the reconstructing framework. Through fusing auxiliary precipitation information, SGD-SM 2.0 products can consider the sudden extreme weather condition for single day in global daily soil moisture products. The comparisons validate the effectiveness of this point in Table 3.

Compared with SGD-SM 1.0, SGD-SM 2.0 outperforms on time-series consistency in Fig. 11(a) and (b). The reconstructed SGD-SM 2.0 points behave more consecutive around their adjacent original soil moistures points than SGD-SM 1.0. While SGD-SM 1.0 exists discrete problem in Fig. 11(a), to some degree. Benefiting from the data fusion of daily precipitation information, the proposed LSTM module can extract time-series features for filling the gaps and missing regions in daily soil moisture products. Though this comparison, the advantage of SGD-SM 2.0 could be better reflected via daily precipitation data fusion and LSTM-CNN model. For daily time-series applications, SGD-SM 2.0 is more suitable than SGD-SM 1.0.

**Q4.4:** *The paper leaves the value of this product relative to the reader as the comparison is made only between versions one and two, and it does not consider the large availability of other soil moisture estimates for multiple uses and applications. Many (hundreds if not thousands) studies currently report alternatives to downscale or fill gaps in satellite soil moisture data. I invite the authors to provide a more extensive literature review and discussion of previous research to support the value of their product.*

**Response:** Thanks for this suggestion. We have provided a more extensive literature review and discussion of previous research to support the value of SGD-SM 2.0 below.

Surface soil moisture acts as a significant part on global hydrology and meteorology, especially for forecasting drought and flood disasters (Wigneron et al., 1999; Long et al., 2014; Brocca et al., 2018). In recent years, satellite-based soil moisture retrieving data has been rapidly progressed on both global and daily monitoring (Shi et al., 2006; Dorigo et al., 2012; Al Bitar et al., 2017; Dorigo et al., 2021). For example, AMSR-E, AMSR2, WindSat global daily soil moisture products and so on (Fan et al., 2004). These quantitative products have been widely utilized for global and long-term hydrological analysis and forecast (Chen et al., 2021; Todd-Brown et al., 2021).

However, because of the limitations of soil moisture retrieving models and satellite orbital covering scopes, the obtained daily soil moisture products are fragmentary and incomplete (Shi et

al., 2002; Enenkel et al, 2016; Meng et al., 2021). As shown in Fig. 1(a) and (b), these soil moisture products exist plenty of gap regions. Actually, the land coverage rate is only approximately 20% to 80% in daily AMSR-E/2 and WindSat quantitative products (Long et al., 2019).

To settle this adverse effect for global soil moisture applications, most of works adopted the temporal averaging operation such as monthly, quarterly, or yearly averaging (Schaffitel et al., 2020; Guevara et al., 2021; Wang et al., 2021). This strategy could usually acquire full-coverage soil moisture products via averaging abundant daily products. Nevertheless, temporal averaging operation is also a two-edged sword. Firstly, it directly replaces daily temporal resolution with low-frequency temporal resolution (Rebel et al., 2012; Long et al., 2020), which greatly lowers the utilization of daily soil moisture products. Secondly, temporal averaging operation disregards the specific spatial distribution of daily products, and neglects the sequential time-series changing characteristic (Zeng et al., 2015; Wang et al., 2021). In other words, monthly, quarterly, or yearly averaging strategy degrades the original characteristics for daily soil moisture products.

To address this issue, Zhang et al. (2021) generated a seamless, global, daily soil moisture (named SGD-SM 1.0) dataset from 2013 to 2019. The spatial resolution is denoted as 0.25° (about 25km). SGD-SM 1.0 relies on the deep spatio-temporal partial convolutional model to fill the gaps or missing regions in daily soil moisture products. Then three validations are performed to verify the reliability of SGD-SM 1.0 products. Relevant quantitative indexes (R, RMSE and MAE) and results demonstrate that SGD-SM 1.0 products can be extended for global, daily and full-coverage soil moisture measurements (Zhang et al., 2021).

**Table 3.** Comparisons between the SGD-SM 1.0 and SGD-SM 2.0 products (from 2013 to 2019) through selected 124 in-situ sites.

| Dataset version | Average evaluation indicators | | | |
|---|---|---|---|---|
| | R | RMSE | ubRMSE | MAE |
| Monthly-Averaging | 0.612 | 0.147 | 0.089 | 0.115 |
| SGD-SM 1.0 | 0.659 | 0.107 | 0.066 | 0.083 |
| SGD-SM 2.0 | **0.688** | **0.094** | **0.058** | **0.077** |

In addition, we also discuss the accuracy of the soil moisture modeled values against SGD-SM 1.0 and monthly-averaging strategy in Table 3, to support the value of their product. Compared with the SGD-SM 1.0 and monthly-averaging, SGD-SM 2.0 can be effectively utilized for global hydrology monitoring analyzing at fine (daily) temporal resolution, rather than the traditional coarse (monthly/yearly) temporal resolution.

**Q4.5:** *I invite the authors to discuss the main implications of accuracy metrics to assess modeled soil moisture values. Can the authors describe the accuracy of the soil moisture sensors used? I invite the authors to use community accepted standards to report errors on soil moisture products, e.g., ubRMSE https://www.sciencedirect.com/science/article/pii/S0034425720301760, and discuss the accuracy of the soil moisture modeled values against other products or gap-filling efforts. A simple demonstration of the new knowledge that users can obtain from the new product would increase substantially the value of this excellent modeling framework applied to soil moisture satellite estimates.*

**Response:** Thanks for this comment. The global accuracy metrics could be contrasted between the original soil moisture products and SGD-SM 2.0 in Table 1. We have added the ubRMSE index in Table 1, which is a frequently-used metric to validate soil moisture products. We also discuss the accuracy of the soil moisture modeled values against SGD-SM 1.0 and monthly-averaging strategy in Table 3. Compared with the SGD-SM 1.0 and monthly-averaging, SGD-SM 2.0 can be effectively utilized for global hydrology monitoring analyzing at fine (daily) temporal resolution. In addition, this reference [2] has been cited in our manuscript for the use of ubRMSE index.

Reference:

[1] Gruber, A., Lannoy, G. De, Albergel, C., et al.: Validation practices for satellite soil moisture retrievals: What are (the) errors?, Remote Sens. Environ., 244, 111806, 2020.

**Table 1.** Comparisons between the original and SGD-SM 2.0 products (from 2002 to 2022) through 124 selected in-situ sites.

| Soil moisture products | Average evaluation indicators | | | |
|:---:|:---:|:---:|:---:|:---:|
| | R | RMSE | ubRMSE | MAE |
| Original | 0.679 | 0.094 | 0.058 | 0.075 |
| SGD-SM 2.0 | 0.672 | 0.096 | 0.061 | 0.078 |

**Table 3.** Comparisons between the SGD-SM 1.0 and SGD-SM 2.0 products (from 2013 to 2019) through selected 124 in-situ sites.

| Dataset version | Average evaluation indicators | | | |
|:---:|:---:|:---:|:---:|:---:|
| | R | RMSE | ubRMSE | MAE |
| Monthly-Averaging | 0.612 | 0.147 | 0.089 | 0.115 |
| SGD-SM 1.0 | 0.659 | 0.107 | 0.066 | 0.083 |
| SGD-SM 2.0 | **0.688** | **0.094** | **0.058** | **0.077** |

**Q4.6:** *Finally, but more importantly in my opinion (considering that this is a dataset journal), please consider publishing your code in order to fulfill the FAIR principles and contribute to the open-science culture transparently e.g., https://bg.copernicus.org/preprints/bg-2021-323/bg-2021-323.pdf..*

**Response:** Thanks for this suggestion. To contribute to the open-science culture, we have published our code at https://github.com/qzhang95/SGD-SM. More subsequent information of this code will be maintained at GitHub. In addition, this reference [2] has been cited in our manuscript.

**Q4.9:** *L40-65 Consider combining each weakness or limitation in v1 with their corresponding advantages in v2, instead of two separated lists.*

**Response:** Thanks for this suggestion. We have combined each weakness or limitation in SGD-SM 1.0 with their corresponding advantages in SGD-SM 2.0 in the revised manuscript:

★ SGD-SM 1.0 only uses single sensor (AMSR2), and the temporal range is insufficient with just seven years. While global soil moisture analysis and applications generally need longer-term and more multi-sensors products. The application range of SGD-SM 1.0 is still limited. Compared with SGD-SM 1.0, SGD-SM 2.0 uses three passive microwave sensors (AMSR-E, WindSat, and AMSR2). Temporal range of SGD-SM 2.0 is extended to twenty years from 2002 to 2022. The application scope of SGD-SM 2.0 could be enlarged through these long-term soil moisture products.

★ SGD-SM 1.0 ignores the daily extreme weather condition. If one day occurs a sudden precipitation, SGD-SM 1.0 usually performs poor under this scenario. The main reason is that SGD-SM 1.0 relies on the internal spatio-temporal correlation, which not considers the external environmental factors. Compared with SGD-SM 1.0, SGD-SM 2.0 introduces the global daily precipitation products into the reconstructing framework. Through fusing auxiliary precipitation data, SGD-SM 2.0 could lead in the daily extreme weather information for gap-filling.

★ Although SGD-SM 1.0 employs 3-D partial convolutional neural network to exploit both spatial and temporal feature, it is still insufficient for utilizing sequential time-series information. For daily soil moisture products, how to effectively reconstruct gaps missing regions through interrelated temporal information is significant. Compared with SGD-SM 1.0, SGD-SM 2.0 develops an integrated long and short-term memory convolutional neural network (LSTM-CNN) to fill the gaps and missing regions in these daily products. The proposed LSTM-CNN model could simultaneously utilize recurrent time-series information and spatial information.

★ Compared with SGD-SM 1.0 products, SGD-SM 2.0 products outperform on R (0.688), RMSE (0.094), and MAE (0.077). In addition, the time-series curves of the improved SGD-SM 2.0 products are more consistency with the original daily time-series soil moisture values. Benefiting from the data fusion of daily precipitation information, the proposed LSTM module can extract time-series features for filling the gaps and missing regions in

daily soil moisture products. Therefore, SGD-SM 2.0 can be effectively utilized for global hydrology monitoring analyzing at fine (daily) temporal resolution.

**Q4.10:** *L83 how they are employed?*

**Response:** Thanks for this query. We have rewritten this sentence as "These recorded AMSR-E, WindSat and AMSR2 global daily products are all employed as the initial input of the proposed LSTM-CNN model for generating SGD-SM 2.0 products." in current manuscript.

**Q4.11:** *L98 What is the criteria to select those sites?*

**Response:** Thanks for this problem. In our in-situ validation, we select 124 sites from ISMN. The selected criteria include three points: 1) The in-situ soil moisture sites are downloadable through the given website. 2) The in-situ soil moisture sites are continuous for the long-term observation, at least one year. 3) The spatial distribution of these in-situ sites covers various continents, land use and soil types. We have supplemented these descriptions in our revised version.

**Q4.12:** *L102 It seems to me that the authors solve a regression problem (where soil moisture is a response of precipitation and time) using deep learning, but they use the word assimilation, which is relatively fine for me given how the algorithm they use works. However I recommend to elaborate on the concept of data assimilation applied here for a better and broader understanding of narrative flow.*

**Response:** Thanks for this suggestion. We also agree that "data assimilation" is generally used optimally combine numerical models with observations. In this work, SGD-SM 2.0 introduces the global daily precipitation products into the reconstructing framework. Through the auxiliary precipitation data, SGD-SM 2.0 could lead in the daily extreme weather information for gap-filling. Therefore, we have replaced "assimilation" as "fusion" in the whole manuscript, to better embody the meaning of multi-source products fusion (precipitation and soil moisture).

**Q4.13:** *L108 what soil moisture product? the authors use three products.*

**Response:** Thanks for this issue. In this sentence, it stands for the arbitrary soil moisture product (AMSR-E, WindSat or AMSR2) for date $T$. We have added this description in current version.

**Q4.14:** *L170 40×40 what?*

**Response:** Thanks for this query. To optimize the proposed LSTM-CNN model, we need to build the training dataset with huge number. This training dataset is composed of lots of spatial patches, which are cropped from the original soil moisture products. 40×40 represents the spatial dimension of these patches in the training dataset.

**Q4.15:** *L219 What was the criteria to select those 124 sites? Can the authors provide a map of points showing in colors the correlation between in-situ and their product for all the stations? I like the presented information but this is a global product and I think it will be useful to interpret the reliability of the product elsewhere. Also for bias indicators (MAE, RMSE), it would be nice to see a map of errors to identify areas with high or low quality of predictions. Please consider*

*also the ubRMSE as it has been a widely discussed metric validating local to global soil moisture*

*predictions. Please discuss the values of accuracy metrics in this and other products.*

**Response:** Thanks for this comment. In our in-situ validation, we select 124 sites from ISMN. The selected criteria include three points: 1) The in-situ soil moisture sites are downloadable through the given website. 2) The in-situ soil moisture sites are continuous for the long-term observation, at least one year. 3) The spatial distribution of these in-situ sites covers various continents, land use and soil types. In terms of the map whose points show in colors the correlation between in-situ and their product for all the stations, the scatter is too time-consuming to depict, due to the huge amount points. Therefore, we give six scatters for single in-situ to reveal the accuracy of SGD-SM 2.0. The global indexes and errors could be contrasted between the original soil moisture products and SGD-SM 2.0 in Table 1. We have added the ubRMSE index in Table 1, which is a frequently-used metric to validate soil moisture products. We also discuss the accuracy of the soil moisture modeled values against SGD-SM 1.0 and monthly-averaging strategy in Table 3. Compared with the SGD-SM 1.0 and monthly-averaging, SGD-SM 2.0 can be effectively utilized for global hydrology monitoring analyzing at fine (daily) temporal resolution.

**Table 1.** Comparisons between the original and SGD-SM 2.0 products (from 2002 to 2022).

| Soil moisture products | Average evaluation indicators | | | |
|---|---|---|---|---|
| | R | RMSE | ubRMSE | MAE |
| Original | 0.679 | 0.094 | 0.058 | 0.075 |
| SGD-SM 2.0 | 0.672 | 0.096 | 0.061 | 0.078 |

**Table 3.** Comparisons between the SGD-SM 1.0 and SGD-SM 2.0 products (from 2013 to 2019).

| Dataset version | Average evaluation indicators | | | |
|---|---|---|---|---|
| | R | RMSE | ubRMSE | MAE |
| Monthly-Averaging | 0.612 | 0.147 | 0.089 | 0.115 |
| SGD-SM 1.0 | 0.659 | 0.107 | 0.066 | 0.083 |
| SGD-SM 2.0 | **0.688** | **0.094** | **0.058** | **0.077** |

**Q4.16:** *L225 can the authors highlight these points in figure 3b?*

**Response:** Thanks for this suggestion. We have highlighted these six in-situ points in Figure 3b (marked as blue blue circles), for the better reading and understanding.

[Figure]

**Figure 3b.** Spatial distribution of selected in-situ data.

**Q4.17:** *L285 the temporal resolution depends on the application.*

**Response:** Thanks for this comment. We have revised this sentence as "Therefore, SGD-SM 2.0 can be effectively utilized for global hydrology monitoring analyzing at fine (daily) temporal resolution." in current manuscript.

**Q4.18:** *Figures 9 and 10, consider using lines instead of points.*

**Response:** Thanks for this suggestion. In Figures 9 and 10, we also considered use lines to reveal the time-series relation in SGD-SM 2.0. Nevertheless, the main purpose in Figures 9 and 10 is to highlight the reconstructed soil moisture values (red points), especially for the time-series relation with the original soil moisture values (blue points). While the line charts cannot ensure this purpose for SGD-SM 2.0. Therefore, we utilize points rather than lines in the time-series validation.

---

## Author Response (AR2)

**Author's Response for All the Comments**

Dear Topical Editor and Referees:

We are particularly grateful for your careful reading, and for giving us many constructive comments of this work!

According to the comments and suggestions, we have tried our best to improve the previous manuscript essd-2022-80 (SGD-SM 2.0: An Improved Seamless Global Daily Soil Moisture Long-term Dataset From 2002 to 2022). The modified words or sentences are marked as blue color in the revised manuscript. An item-by-item response follows.

Once again, we are particularly grateful for your careful reading and constructive comments. Thanks very much for your time.

Best regards,

Qiang Zhang and all co-authors

**Response to the Comments of Topical Editor:**

**General comments:**

> *Please, revise your manuscript by considering the suggestions provided by one of the Referees.*

**Response:** We are particularly grateful to the editor and reviewers for these detailed suggestions! Based on the reviewers' comments, we have tried our best to revise the previous manuscript. An item-by-item response to each constructive comment follows.

**Response to the Comments of Reviewer #3:**

*Many improvements have been made to the manuscript. However, two important issues still need to be resolved.*

**Response:** We are particularly grateful to the reviewer for his/her detailed suggestion! According to the comments, we have tried our best to improve the previous manuscript. An item-by-item response to each constructive comment follows.

**Major comments:**

**Q1.1:** *First issue: In my first review, I indicated that some soil moisture values shown in Fig. 8 show extremely SM high values of more than 80 Vol.%. Such high values are very unlikely, as soil porosity in most soil is typically between 40-50 Vol.%.*

*The authors responded:*

*"Actually, the SM values in this work are the volume ratio (unit: m3m-3, from 0% to 100%), rather than the mass ratio (kgm-3, usually 0% to 50%). This phenomenon is normal because of the unit via volume ratio, not measurement errors or SM overestimation by the CNN. (...) Overall, these outliers are few with small impact for SGD-SM 2.0. Therefore, we don't clean the data with an outlier detection method."*

*In fact, I was also referring to the volume ratio. Again, SM values above 80 Vol.% (i.e. above 0.8 m3m-3) are unrealistic. The authors also replied that there are only few outliers where*

*SM is above the porosity of the soil. However, this could be a bigger problem that definitely needs to be better addressed to prevent users from using unrealistic SM values for their analyses. I suggest that the authors identify outliers by comparing the SM data product with the porosity information from the global soil database SoilGrid (https://www.isric.org/explore/soilgrids). This would be an easy-to-implement outlier detection.*

**Response:** Thanks for this comment. We also agree that SM values above 80 Vol.% (i.e. above 0.8 $m^3 \cdot m^{-3}$) are unrealistic. Nevertheless, one of the limitations for SGD-SM 2.0 is that the proposed reconstructing framework fully relies on the initial satellite-based SM information. Even if the original SM values are above 80 Vol.%, the proposed reconstructing framework still take them as the valid SM information for gap-filling. These retrieving errors (i.e. above 0.8 $m^3 \cdot m^{-3}$) in the initial satellite-based SM data are also inevitably transmitted into the SGD-SM 2.0 products. This limitation has been supplemented into the current discussion below:

"1) The errors of original AMSR-E/WindSat/AMSR2 products: The proposed SGD-SM product is generated based on original AMSR-E/WindSat/AMSR2 products. While these passive soil moisture products also exist errors (i.e. above 0.8 $m^3 \cdot m^{-3}$) , due to the satellite sensor imaging and soil moisture retrieval algorithm. As shown in Table 1, the R, RMSE, and MAE evaluation indexes of the original products are 0.679, 0.094, and 0.075, respectively. These errors are also inevitably transmitted into the generated SGD-SM 2.0 products. In other words, SGD-SM 2.0 absolutely trusts the initial satellite-based SM values without any hesitation."

In our future work (SGD-SM 3.0), we will introduce the outlier filtering strategy, to exclude these initial SM exception information (above 80 Vol.%). Identifying outliers by comparing the SM data product with the porosity information from the global soil database SoilGrid will also be utilized in SGD-SM 3.0.

**Q1.2:** *Second issue: The authors now use data six stations from Bogena et al. (2022), but these are not part of ISMN database. The in-situ soil moisture data from ISMN used in this work are still treated anonymously. However, this practice is not in accordance with the General Terms and Conditions (https://ismn.geo.tuwien.ac.at/en/terms-and-conditions/), where it is written: "Whenever data distributed by the ISMN are being used for publication, the data's origin (i.e. the original data provider and the ISMN) must be acknowledged and referenced. A reference both to the ISMN AND to all networks providing data for the study in question shall be given." Therefore, the authors still have to add at table with basic information on the soil moisture data using, including the name of the site owners and/or monitoring networks.*

**Response:** Thanks for this suggestion. In this work, we select 124 stations from ISMN from 2002 to 2022 and match them with corresponding soil moisture product in SGD-SM 2.0. In other words, all the selected 124 in-situ sites (including COSMOS, SD-DEM, SMOS-CBR, SCAN, PBO-H2O, USCRN and OZNET networks) are employed to validate the accuracy of SGD-SM 2.0. for better scatter visualization, we chose partial in-situ soil moisture stations as examples. In addition, we have added acknowledgments and references of ISMN in the revised manuscript below:

Acknowledgments: We appreciatively acknowledge GES DISC and ISMN, for them releasing related products and in-situ sites.

References:

[1] International Soil Moisture Network (ISMN): https://ismn.geo.tuwien.ac.at/en/.

[2] Dorigo, W. A., Wagner, W., Hohensinn, R., Hahn, S., Paulik, C., Xaver, A., Gruber, A., Drusch, M., Mecklenburg, S., van Oevelen, P., Robock, A., and Jackson, T.: The International Soil Moisture Network: a data hosting facility for global in situ soil moisture measurements, Hydrol. Earth Syst. Sci., 15, 1675–1698, https://doi.org/10.5194/hess-15-1675-2011, 2011.

[3] Dorigo, W. A., Xaver, A., Vreugdenhil, M., Gruber, A., Hegyiová, A., Sanchis-Dufau, A. D., Zamojski, D., Cordes, C., Wagner, W., and Drusch, M.: Global automated quality control of in situ soil moisture data from the international soil moisture network, Vadose Zone J., 12, 1–21, https://doi.org/10.2136/vzj2012.0097, 2013.

[4] Dorigo, W., Himmelbauer, I., Aberer, D. et al.: The International Soil Moisture Network: serving Earth system science for over a decade, Hydrol. Earth Syst. Sci., 25, 5749–5804, https://doi.org/10.5194/hess-25-5749-2021, 2021.

**Table 2.** 124 selected soil moisture stations from ISMN from 2002 to 2022 for validating SGD-SM 2.0.

| COSMOS-001 | COSMOS-004 | COSMOS-006 | COSMOS-007 | COSMOS-010 | COSMOS-011 |
|---|---|---|---|---|---|
| COSMOS-012 | COSMOS-013 | COSMOS-014 | COSMOS-015 | COSMOS-016 | COSMOS-017 |
| COSMOS-018 | COSMOS-020 | COSMOS-021 | COSMOS-023 | COSMOS-024 | COSMOS-025 |
| COSMOS-026 | COSMOS-027 | COSMOS-028 | COSMOS-029 | COSMOS-030 | COSMOS-031 |
| COSMOS-032 | COSMOS-033 | COSMOS-034 | COSMOS-035 | COSMOS-038 | COSMOS-039 |
| COSMOS-040 | COSMOS-041 | COSMOS-042 | COSMOS-043 | COSMOS-044 | COSMOS-045 |
| COSMOS-046 | COSMOS-047 | COSMOS-048 | COSMOS-049 | COSMOS-050 | COSMOS-051 |
| COSMOS-052 | COSMOS-053 | COSMOS-054 | COSMOS-055 | COSMOS-056 | COSMOS-057 |
| COSMOS-058 | COSMOS-060 | COSMOS-061 | COSMOS-062 | COSMOS-063 | COSMOS-064 |
| COSMOS-066 | COSMOS-067 | COSMOS-068 | COSMOS-069 | COSMOS-070 | COSMOS-071 |
| COSMOS-072 | COSMOS-073 | COSMOS-074 | COSMOS-076 | COSMOS-078 | COSMOS-081 |
| COSMOS-082 | COSMOS-084 | COSMOS-087 | COSMOS-089 | COSMOS-090 | COSMOS-091 |
| COSMOS-092 | COSMOS-093 | COSMOS-094 | COSMOS-095 | COSMOS-096 | COSMOS-098 |
| COSMOS-099 | COSMOS-100 | COSMOS-101 | COSMOS-102 | COSMOS-103 | COSMOS-105 |
| COSMOS-107 | COSMOS-108 | COSMOS-109 | COSMOS-110 | COSMOS-111 | COSMOS-123 |
| RSMN-15136 | RSMN-15199 | RSMN-15412 | RSMN-15470 | RSMN-15479 | SD-DEM |
| SMOS-CBR | SMOS-LHS | SMOS-MTM | SMOS-SFL | SMOS-SVN | SMOS-PZN |
| SCAN-2014 | SCAN-2046 | SCAN-2055 | SCAN-2087 | SCAN-2179 | SCAN-2181 |
| PBO-076 | PBO-094 | PBO-250 | PBO-472 | PBO-474 | PBO-482 |
| PBO-498 | PBO-508 | PBO-525 | PBO-569 | PBO-742 | PBO-811 |
| USCRN-011 | USCRN-020 | OZNET-K1 | OZNET-K2 | | |

These 124 selected soil moisture stations from ISMN from 2002 to 2022 are shown in Table 2, for validating SGD-SM 2.0. Besides, we have also added a table with basic information on the

in-situ soil moisture sites like Bogena et al. (2022). As listed in Table 3, it includes the name of the station, country, longitude/latitude, main land use, lattice water, and soil organic carbon. Due to the page limiting of this manuscript (it is too long to show the basic information of 124 selected sites...), we give representative in-situ sites (Taking partial sites as example, including COSMOS, SD-DEM, SMOS-CBR, SCAN, PBO, USCRN and OZNET networks) in Table 3 as follow:

**Table 3.** Basic information on the in-situ soil moisture sites (Taking partial sites as examples).

| Station | Lon/Lat | Elevation (m) | main land use | lattice water | soil organic carbon |
|---------|---------|---------------|---------------|---------------|---------------------|
| COSMOS-016 | 42.537, -72.171 | 316 | Crop | 4.50% | 1.59% |
| COSMOS-055 | 0.282, 36.866 | 1824 | Bush | 6.10% | 1.11% |
| COSMOS-082 | 48.141, 15.171 | 73 | Grass | 2.10% | 1.93% |
| COSMOS-096 | -14.159, 131.388 | 169 | Silty Sand | 2.30% | 1.24% |
| COSMOS-101 | -21.617, -47.632 | 563 | Grass | 1.70% | 1.87% |
| COSMOS-123 | 31.369, 91.899 | 1201 | Forest | 4.40% | 2.36% |
| SD-DEM | 13.287, 30.479 | 864 | Coarse Sand | 1.30% | 0.98% |
| SMOS-CBR | 42.563, 13.798 | 52 | Grass | 3.40% | 2.25% |
| SCAN-2014 | 38.173, -65.171 | 274 | Crop | 2.20% | 1.97% |
| PBO-076 | 24.189, -81.343 | 156 | Silty Sand | 1.90% | 1.14% |
| USCRN-011 | 20.507, -97.662 | 583 | Grass | 3.70% | 1.98% |
| OZNET-K1 | -21.683, 139.841 | 659 | Scrub | 3.60% | 2.34% |